# An Online Sequential Test for Qualitative Treatment Effects

## Abstract

Tech companies (e.g., Google or Facebook) often use randomized online experiments and/or A/B testing primarily based on the average treatment effects to compare their new product with an old one. However, it is also critically important to detect qualitative treatment effects such that the new one may significantly outperform the existing one only under some specific circumstances. The aim of this paper is to develop a powerful testing procedure to efficiently detect such qualitative treatment effects. We propose a scalable online updating algorithm to implement our test procedure. It has three novelties including adaptive randomization, sequential monitoring, and online updating with guaranteed type-I error control. We also thoroughly examine the theoretical properties of our testing procedure including the limiting distribution of test statistics and the justification of an efficient bootstrap method. Extensive empirical studies are conducted to examine the finite sample performance of our test procedure.

## 1 Introduction

Tech companies use randomized online experiments, or A/B testing to compare their new product with a well-established one. Most works in the literature focus on the average treatment effects (ATE) between the new and existing products (see Kharitonov et al., 2015; Johari et al., 2015; 2017; Yang et al., 2017; Ju et al., 2019, and the references therein). In addition to ATE, sometimes we are interested in locating the subgroup (if exists) that the new product performs significantly better than the existing one, as early as possible. Consider a ride-hailing company (e.g., Uber). Suppose some passengers are in the recession state (at a high risk of stopping using the companys app) and the company comes up with certain strategy to intervene the recession process. We would like to if there are some subgroups that are sensitive to the strategy and pin-point these subgroups if exists. It motivates us to consider the null hypothesis that the treatment effect is nonpositive for all passenger.

Such a null hypothesis is closely related to the notion of qualitative treatment effects in medical studies (QTE, Gail & Simon, 1985; Roth & Simon, 2018; Shi et al., 2020a), and conditional moment inequalities in economics (see for example, Andrews & Shi, 2013; 2014; Chernozhukov et al., 2013; Armstrong & Chan, 2016; Chang et al., 2015; Hsu, 2017). However, these tests are computed offline and might not be suitable to implement in online settings. Moreover, it is assumed in those papers that observations are independent. In online experiment, one may wish to adaptively allocate the treatment based on the observed data stream in order to maximize the cumulative reward or to detect the alternative more efficiently. The independence assumption is thus violated. In addition, an online experiment is desired to be terminated as early as possible in order to save time and budget. Sequential testing for qualitative treatment effects has been less explored.

In the literature, there is a line of research on estimation and inference of the heterogeneous treatment effects (HTE) (Athey & Imbens, 2016; Taddy et al., 2016; Wager & Athey, 2018; Yu et al., 2020). In particular, Yu et al. (2020) proposed an online test for HTE. We remark that HTE and QTE are related yet fundamentally different hypotheses. There are cases where HTE exists whereas QTE does not. See Figure 1 for an illustration. Consequently, applying their test will fail in our setting.

The contributions of this paper are summarized as follows. First, we propose a new testing procedure for treatment comparison based on the notion of QTE. When the null hypothesis is not rejected, the new product is no better than the control for any realization of covariates, and thus it is not useful at all. Otherwise, the company could implement different products according to the auxiliary

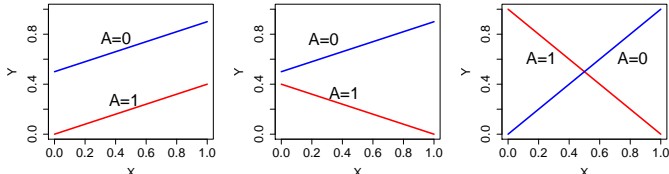

Figure 1: Plots demonstrating QTE. $X$ denotes the observed covariates, $A$ denotes the received treatment and $Y$ denotes the associated reward. In the ride-hailing example, $X$ is a feature vector describing the characteristics of a passenger, $A$ is a binary strategy indicator and $Y$ is the passenger's number of rides in the following two weeks. In the left panel, the treatment effect does not depend on $X$. Neither HTE nor QTE exists in this case. In the middle panel, HTE exists. However, the treatment effect is always negative. As such, QTE does not exist. In the right penal, both QTE and HTE exist.

covariates observed, to maximize the average reward obtained. We remark that there are plenty cases where the treatment effects are always nonpositive (see Section 5 of Chang et al., 2015; Shi et al., 2020a). A by-product of our test is that it yields a decision rule to implement personalization when the null is rejected (see Section 3.1 for details). Although we primarily focus on QTE in this paper, our procedure can be easily extended to testing ATE as well (see Appendix D for details).

Second, we propose a scalable online updating algorithm to implement our test. To allow for sequential monitoring, our procedure leverages idea from the $\alpha$ spending function approach (Lan & DeMets, 1983) originally designed for sequential analysis in a clinical trial (see Jennison & Turnbull, 1999, for an overview). Classical sequential tests focus on ATE. The test statistic at each interim stage is asymptotically normal and the stopping boundary can be recursively updated via numerical integration. However, the limiting distribution of the proposed test statistic does not have a tractable analytical form, making the numerical integration method difficult to apply. To resolve this issue, we propose a scalable bootstrap-assisted procedure to determine the stopping boundary.

Third, we adopt a theoretical framework that allows the maximum number of interim analyses $K$ to diverge as the number of observations increases, since tech companies might analyze the results every few minutes (or hours) to determine whether to stop the experiment or continue collecting more data. It is ultimately different from classical sequential analysis where $K$ is fixed. Moreover, the derivation of the asymptotic property of the proposed test is further complicated due to the adaptive randomization procedure, which makes observations dependent of each other. Despite these technical challenges, we establish a nonasymptotic upper bound on the type-I error rate by explicitly characterizing the conditions needed on randomization procedure, $K$ and the number of samples observed at the initial decision point to ensure the validity of our test.

## 2 BACKGROUND AND PROBLEM FORMULATION

We propose a potential outcome framework (Rubin, 2005) to formulate our problem. Suppose that we have two products including the control and the treatment. The observed data at time point $t$ consists of a sequence of triples $\{(X_i, A_i, Y_i)\}_{i=1}^{N(t)}$, where $N(\cdot)$ is a counting process that is independent of the data stream $\{(X_i, A_i, Y_i)\}_{i=1}^{+\infty}$, $A_i$ is a binary random variable indicating the product executed for the $i$-th experiment, $X_i \in \mathbb{R}^p$ denotes the associated covariates, and $Y_i$ stands for the associated reward (the larger the better by convention). We allow $A_i$ to depend on $X_i$ and past observations $\{(X_j, A_j, Y_j)\}_{j<i}$ so that the randomization procedure can be adaptively changed. In addition, define $Y_i^*(0)$ and $Y_i^*(1)$ to be the potential outcome that would have been observed if the corresponding product is executed for the $i$-th experiment. Suppose that $\{(X_i, Y_i^*(0), Y_i^*(1))\}_{i=1}^{+\infty}$ are independently and identically distributed copies of $(X, Y^*(0), Y^*(1))$. Let $\mathbb{X}$ be the support of $X$ and $Q_0(x, a) = \mathrm{E}\{Y^*(a)|X = x\}$ for $a = 0, 1$, we focus on testing the following hypotheses:

$$H_0 : Q_0(x, 1) \leq Q_0(x, 0), \forall x \in \mathbb{X} \quad \text{versus} \quad H_1 : Q_0(x, 1) > Q_0(x, 0), \exists x \in \mathbb{X}.$$

Notice that when there are no covariates, i.e., $\mathbb{X} = \emptyset$, the hypotheses are reduced to $H_0 : \tau_0 \leq 0$ versus $H_1 : \tau_0 > 0$, where $\tau_0$ corresponds to ATE, i.e, $\tau_0 = \mathrm{E}\{Y^*(1) - Y^*(0)\}$. In general, we require $\mathbb{X}$ to be a compact set. We consider a large linear approximation space $\mathcal{Q}$ for the conditional mean function $Q_0$. Specifically, let $\mathcal{Q} = \{Q(x, a; \beta_0, \beta_1) = \varphi^\top(x)\beta_a : \beta_0, \beta_1 \in \mathbb{R}^q\}$ be the approximation space, where $\varphi(x)$ is a $q$-dimensional vector composed of basis functions on $\mathbb{X}$. The

dimension $q$ is allowed to diverge with the number of observations in order to alleviate the effects of model misspecification. The use of linear approximation space simplifies the computation of our testing procedure. When $Q_0$ is well approximated, it suffices to test

$$H_0 : \varphi^\top(x)(\beta_1^* - \beta_0^*) \leq 0, \forall x \in \mathbb{X} \quad \text{versus} \quad H_1 : \varphi^\top(x)(\beta_1^* - \beta_0^*) > 0, \exists x \in \mathbb{X}. \tag{1}$$

For clarity, here we assume $Q_0(x, a) = Q(x, a; \beta_0^*, \beta_1^*)$ for some $\beta_0^*$ and $\beta_1^*$. In Appendix B, we allow the approximation error $\inf_{\beta_0, \beta_1 \in \mathbb{R}^p} \sup_{x \in \mathbb{X}, a \in \{0,1\}} |Q_0(x, a) - Q(x, a; \beta_0, \beta_1)|$ to be nonzero.

Let $\mathcal{F}_j$ denote the sub-dataset $\{(X_i, A_i, Y_i)\}_{1 \leq i \leq j}$ for $j \geq 1$ and $\mathcal{F}_0 = \emptyset$. Throughout this paper, we assume that the following two assumptions hold.

(A1) $Y_i = A_i Y_i^*(1) + (1 - A_i) Y_i^*(0)$ for $\forall i \geq 1$.
(A2) $A_i$ is independent of $Y_i^*(0), Y_i^*(1), \{(X_k, Y_k^*(0), Y_k^*(1))\}_{k > i}$ given $X_i$ and $\mathcal{F}_{i-1}$, for any $i$.

Assumption (A1) is referred to be the stable unit treatment value assumption (Rubin, 1974) and Assumption (A2) is the sequential randomization assumption (Zhang et al., 2013) and is automatically satisfied in a randomized study where the treatments are independently generated of the observed data. (A2) essentially assumes there is no unmeasured confounders. These assumptions guarantee that both regression coefficients (defined through potential outcomes) are estimable from the observed dataset as shown in the following lemma.

**Lemma 1** *Let $\mathbb{I}(\cdot)$ denotes the indicator function. Under (A1)-(A2), we have*

$$E[\mathbb{I}(A_i = a)\{Y_i - \varphi^\top(X_i)\beta_a^*\}] = 0, \quad \forall a \in \{0, 1\}, i \geq 1.$$

## 3 Online sequential testing for QTE

### 3.1 Test statistics and their limiting distribution

We first present our test statistic for testing $H_0$. In view of Lemma 1, we estimate $\beta_a$ by using the ordinary least squares estimator

$$\widehat{\beta}_a(t) = \widehat{\Sigma}_a^{-1}(t) \left\{ \frac{1}{N(t)} \sum_{i=1}^{N(t)} \mathbb{I}(A_i = a)\varphi(X_i)Y_i \right\}$$

at each time point $t$ for $a \in \{0, 1\}$, where $\widehat{\Sigma}_a(t) = N^{-1}(t) \sum_{i=1}^{N(t)} \mathbb{I}(A_i = a)\varphi(X_i)\varphi^\top(X_i)$. A generalized inverse might be used even if $\widehat{\Sigma}_a(t)$ is not invertible. Consider the following test statistic $S(t) = \sup_{x \in \mathbb{X}} \varphi^\top(x)\{\widehat{\beta}_1(t) - \widehat{\beta}_0(t)\}$. Under $H_0$, we expect $S(t)$ to be small. A large $S(t)$ can be interpreted as the evidence against $H_0$. As such, we reject $H_0$ for large $S(t)$. We remark that when $H_0$ is rejected, we can apply the decision rule $d(x) = \arg\max_{a \in \{0,1\}} \varphi^\top(x)\widehat{\beta}_a(t)$ for personalized recommendation.

To determine the rejection region, we next discuss the limiting distribution of $S(t)$. Under $H_0$,

$$S(t) \leq \sup_{x \in \mathbb{X}} \varphi^\top(x)\{\widehat{\beta}_1(t) - \beta_1^* - \widehat{\beta}_0(t) + \beta_0^*\} + \sup_{x \in \mathbb{X}} \varphi^\top(x)(\beta_1^* - \beta_0^*)$$

$$\leq \sup_{x \in \mathbb{X}} \varphi^\top(x)\{\widehat{\beta}_1(t) - \beta_1^* - \widehat{\beta}_0(t) + \beta_0^*\}. \tag{2}$$

Both equalities hold when $\beta_0^* = \beta_1^*$. Suppose there exists some function $\pi^*(\cdot, \cdot)$ defined on $\{0, 1\} \times \mathbb{X}$ that satisfies $\mathrm{E}^X |\sum_{i=1}^n n^{-1}\pi_{i-1}(a, X) - \pi^*(a, X)| \overset{P}{\to} 0, \forall a \in \{0, 1\}$ as $n \to \infty$, where $\pi_n(\cdot, \cdot) = \Pr(A_n = a | X_n = x, \mathcal{F}_{n-1})$, and the expectation $\mathrm{E}^X$ is taken with respect to $X$. This condition implies that the treatment assignment mechanism cannot be arbitrary (see the discussion below Theorem 1 for details). Then we will show

$$B(t) \equiv \sqrt{N(t)}\{\widehat{\beta}_1(t) - \beta_1^* - \widehat{\beta}_0(t) + \beta_0^*\} \overset{d}{\to} N\left(0, \sum_{a \in \{0,1\}} \Sigma_a^{-1} \Phi_a \Sigma_a^{-1}\right), \quad \text{as } N(t) \to \infty, \tag{3}$$

where $\Sigma_a = \mathrm{E}\pi^*(a, X)\varphi(X)\varphi^\top(X)$, $\Phi_a = \mathrm{E}\pi^*(a, X)\sigma^2(a, X)\varphi(X)\varphi^\top(X)$, and $\sigma^2(a, x) = \mathrm{E}[\{Y^*(a) - \varphi^\top(X)\beta_a\}^2 | X = x]$, for any $x \in \mathbb{X}$. According to equation 3, the right-hand-side

(RHS) of equation 2 is to converge in distribution to the maximum of some Gaussian random variables. This observation forms the basis of our test.

We next discuss the sequential implementation of our test. Assume that the interim analyses are conducted at time points $t_1, t_2, \ldots, t_K \in [0, \ldots, T]$ such that $0 < t_1 < t_2 < \cdots < t_K = T$. We allow $K$ to grow with the number of observations. In the most extreme case, one may set $t_k = \inf_t \{N(t) \geq N(t_{k-1})+1\}, \forall k \geq 2$. That is, we make a decision regarding the null hypothesis upon the arrival of each observation. In addition, we assume that $t_1$ is large so that there are enough number of samples $N(t_1)$ to guarantee the validity of the normal approximation for $B(t_1)$. We remark that in typical tech companies such as Amazon, Facebook, etc., massive data are collected even within a short time interval. Large sample approximation is validated in these applications.

To guarantee our test controls the type-I error, we reject $H_0$ and terminate the experiment at $t_k$ if $\sqrt{N(t_k)}S(t_k) \geq z_k$ for some $k = 1, \ldots, K$ with some suitably chosen $z_1, \ldots, z_K > 0$ that satisfy

$$\Pr\left(\max_{k \in \{1,\ldots,K\}}\{\sqrt{N(t_k)}S(t_k) - z_k\} > 0\right) \leq \alpha + o(1)$$

for a given significance level $\alpha > 0$ under $H_0$. In view of equation 2, it suffices to find $\{z_k\}_k$ that satisfy

$$\Pr\left\{\max_{k \in \{1,\ldots,K\}}\left(\sup_{x \in \mathbb{X}} \varphi^\top(x)B(t_k) - z_k\right) > 0\right\} \leq \alpha + o(1), \tag{4}$$

where the stochastic process $B(\cdot)$ is defined in equation 3.

To determine $\{z_k\}_k$, we need to derive the asymptotic distribution of the left-hand-side (LHS) of equation 4. To this end, define a mean-zero Gaussian process $G(t)$ with covariance function

$$\mathrm{Cov}(G(t), G(t')) = N^{1/2}(t)N^{-1/2}(t') \sum_{a \in \{0,1\}} \Sigma_a^{-1}\Phi_a\Sigma_a^{-1}, \quad \forall 0 < t \leq t'.$$

In the following, we show that the LHS of equation 4 can be uniformly approximated by $G(\cdot)$, for any $\{z_k\}_{k=1,\ldots,K}$. To establish our theoretical results, we need some regularity conditions on $\varphi(\cdot)$. To save space, we summarize these assumptions in (A3) and put them in Appendix B.

**Theorem 1** *Assume (A1)-(A3) hold. For $a = 0, 1$, assume $\inf_{x \in \mathbb{X}} \pi^*(a, x) > 0$ and $|Y^*(a)|$ is bounded almost surely. Assume there exists some $0 < \alpha_0 \leq 1$ such that for any sequence $\{j_n\}_n$ that satisfies $j_n^{\alpha_0}/\log^{\alpha_0} j_n \gg q^2$, the following event occurs with probability at least $1 - O(j_n^{-\alpha_0})$,*

$$\sup_{a \in \{0,1\}} E\left|\sum_{i=1}^{k}\{\pi_{i-1}(a, x) - \pi^*(a, x)\}\right| \leq O(1)qk^{1-\alpha_0}\log^{\alpha_0} k, \quad \forall k \geq j_n, \tag{5}$$

*where $O(1)$ denotes some positive constant. Assume $N^{\alpha_0}(t_1)/\log^{\alpha_0} N(t_1) \gg q^2$ and $N(t_1) \gg \log N(T)$ almost surely. Then conditional on the counting process $N(\cdot)$, there exists some constant $c > 0$ such that*

$$\sup_{z_1,\ldots,z_K}\left|\Pr\left\{\max_{k \in \{1,\ldots,K\}}\left(\sup_{x \in \mathbb{X}} \varphi^\top(x)B(t_K) - z_k\right) > 0\right\} - \Pr\left\{\max_{k \in \{1,\ldots,K\}}\left(\sup_{x \in \mathbb{X}} \varphi^\top(x)G(t_K) - z_k\right) > 0\right\}\right|$$
$$\leq c\left[q^{3/4}N^{-1/8}(t_1)\log^{15/8}\{KN(t_1)\} + qN^{-\alpha_0/3}(t_1)\log^{(5+\alpha_0)/3}\{KN(t_1)\}\right].$$

Theorem 1 implies that the approximation error depends on the number of observations obtained up to the first decision point $N(t_1)$, the maximum number of interim analyses $K$, the total number of basis functions $q$, and $\alpha_0$, which characterizes the convergence rate of the treatment assignment mechanism $\sum_{i=1}^{n} n^{-1}\pi_{i-1}$. Clearly, the error will decay to zero when the followings hold with probability tending to 1,

$$q = O(N^{\alpha_*}(t_1)), \quad \text{for some } 0 \leq \alpha^* < \min(1/6, \alpha_0/3), \tag{6}$$

$$\log(K) \ll \min\{N^{1/15-2\alpha^*/5}(t_1), N^{(\alpha_0-3\alpha^*)/(5+\alpha_0)}(t_1)\}. \tag{7}$$

In Appendix C, we show that $\alpha_0 = 1/2$, when an $\epsilon$-greedy strategy is used for randomization to balance the trade-off between exploration and exploitation. In this case, 6 requires $q$ to grow at a slower rate than $N^{1/6}(t_1)$. This condition is automatically satisfied when $q$ is bounded. Condition 7 is satisfied when $K$ grows polynomially fast with respect to $N(t_1)$. In addition to $\epsilon$-greedy, other adaptive allocation procedures (e.g., upper confidence bound or Thompson sampling) could be applied as well.

As discussed in the introduction, the derivation of Theorem 1 is nontrivial. One way to obtain the magnitude of the approximation error is to apply the strong approximation theorem for multidimensional martingales (see Morrow & Philipp, 1982; Zhang, 2004). However, the rate of approximation typically depends on the dimension and decays fast as the dimension increases. To derive Theorem 1, we view $\{\varphi^\top(x)B(t_K)\}_{x\in\mathbb{X}, k\in\{1,...,\kappa\}}$ as a high-dimensional martingale and adopt the Gaussian approximation techniques that have been recently developed by Belloni & Oliveira (2018). In view of equation 2, an application of Theorem 1 yields the following result.

**Theorem 2** *Assume that the conditions of Theorem 1 hold, equation 6 and equation 7 hold with probability tending to 1. Then for any $z_1, \ldots, z_k$ that satisfy*

$$Pr\left\{\max_{k\in\{1,...,K\}}\left(\sup_{x\in\mathbb{X}}\varphi^\top(x)G(t_k) - z_k\right) > 0\right\} = \alpha + o(1), \tag{8}$$

*as $N(t_1)$ diverges to infinity, we have under $H_0$,*

$$Pr\left(\max_{k\in\{1,...,K\}}\{\sqrt{N(t_k)}S(t_k) - z_k\} > 0\right) \leq \alpha + o(1).$$

*The above equality holds when $\beta_0^* = \beta_1^*$.*

Theorem 2 suggests that the type-I error rate of the proposed test can be well controlled. It remains to find critical values $\{z_k\}_{1\leq k\leq K}$ that satisfy equation 8. In the next section, we propose a bootstrap-assisted procedure to determine these critical values.

### 3.2 BOOTSTRAP STOPPING BOUNDARY

We first outline a method based on the wild bootstrap (Wu, 1986) to approximate the limiting distribution of $\{S(t_k)\}_k$. Then we discuss its limitation and present our proposal, a scalable bootstrap algorithm to determine the stopping boundary.

The idea is to generate bootstrap samples $\{\widehat{\beta}_a^{\mathrm{MB}}(t_k)\}_{a,k}$ that have asymptotically the same joint distribution as $\{\widehat{\beta}_a(t_k) - \beta_a^*\}_{a,k}$. Then the joint distribution of $\{S(t_k)\}_k$ can be well-approximated by the conditional distribution of $\{\widehat{S}^{\mathrm{MB}}(t_k)\}_k$ given the data, where $\widehat{S}^{\mathrm{MB}}(t) = \sup_{x\in\mathbb{X}}\varphi^\top(x)\{\widehat{\beta}_1^{\mathrm{MB}}(t) - \widehat{\beta}_0^{\mathrm{MB}}(t)\}$ for any $t$. Specifically, let $\{\xi_i\}_{i=1}^{+\infty}$ be a sequence of i.i.d. standard normal random variables independent of $\{(X_i, A_i, Y_i)\}_{i=1}^{+\infty}$. For $a \in \{0, 1\}$, define

$$\widehat{\beta}_a^{\mathrm{MB}}(t) = \widehat{\Sigma}_a^{-1}(t)\left[\frac{1}{N(t)}\sum_{i=1}^{N(t)}\mathbb{I}(A_i = a)\varphi_i(X)\{Y_i - \varphi^\top(X_i)\widehat{\beta}(t)\}\xi_i\right], \quad \forall a \in \{0, 1\}.$$

Both the asymptotic means of $\sqrt{N(t)}\widehat{\beta}_a^{\mathrm{MB}}(t)$ and $\sqrt{N(t)}(\widehat{\beta}_a(t) - \beta_a^*)$ are zero. In addition, their covariance functions are asymptotically the same. By design, $\{\widehat{\beta}_a^{\mathrm{MB}}(t_k)\}_{a,k}$ is multivariate normal. Similar to equation 3, we can show $\{\widehat{\beta}_a(t_k) - \beta_a^*\}_{a,k}$ is asymptotically multivariate normal. Consequently, the limiting distributions of $\{\widehat{\beta}_a^{\mathrm{MB}}(t_k)\}_{a,k}$ and $\{\widehat{\beta}_a(t_k) - \beta_a^*\}_{a,k}$ are asymptotically equivalent. As such, the bootstrap approximation is valid.

However, calculating $\widehat{\beta}_a^{\mathrm{MB}}(t_k)$ requires $O(N(t_k))$ operations. The time complexity of the resulting bootstrap algorithm is $O(BN(t_k))$ up to the $k$-th interim stage, where $B$ is the total number of bootstrap samples. This can be time consuming when $\{N(t_k) - N(t_{k-1})\}_{k=1}^K$ are large. To facilitate the computation, we observe that in the calculation of $\widehat{\beta}_a^{\mathrm{MB}}$, the random noise is generated upon the arrival of each observation. This is unnecessary as we aim to approximate the distribution of $\widehat{\beta}_a(\cdot)$ only at finitely many time points.

We next present our proposal. Let $\{e_{i,a}\}_{i=1,\ldots,K,a=0,1}$ be a sequence of i.i.d $N(0, I_q)$ random vectors independent of the observed data, where $I_q$ denotes the $q \times q$ identity matrix. At the $k$-th interim stage, we compute $\widehat{S}^{\text{MB}*}(t_k) = \sup_{x \in \mathbb{X}} \varphi^\top(x)\{\widehat{\beta}_1^{\text{MB}*}(t_k) - \widehat{\beta}_0^{\text{MB}*}(t_k)\}$, where $\widehat{\beta}_a^{\text{MB}*}(t_k)$ equals

$$\frac{1}{N(t_k)} \sum_{j=1}^{k} \left( \sum_{i=N(t_{j-1})+1}^{N(t_j)} \widehat{\Sigma}_a^{-1}(t_j)\mathbb{I}(A_i = a)\varphi(X_i)\varphi^\top(X_i)\{Y_i - \varphi(X_i)^\top\widehat{\beta}_a(t_j)\}^2\widehat{\Sigma}_a^{-1}(t_j) \right)^{1/2} e_{j,a}.$$

For any $k_1$ and $k_2$, the conditional covariance of $\sqrt{N(t_{k_1})}\{\widehat{\beta}_1^{\text{MB}*}(t_{k_1}) - \widehat{\beta}_0^{\text{MB}*}(t_{k_1})\}$ and $\sqrt{N(t_{k_2})}\{\widehat{\beta}_1^{\text{MB}*}(t_{k_2}) - \widehat{\beta}_0^{\text{MB}*}(t_{k_2})\}$ equals

$$\frac{1}{\sqrt{N(t_{k_1})N(t_{k_2})}} \sum_{a=0}^{1} \sum_{j=1}^{k_1} \sum_{i=N(t_{j-1})+1}^{N(t_j)} \widehat{\Sigma}_a^{-1}(t_j)\mathbb{I}(A_i = a)\varphi(X_i)\varphi^\top(X_i)\{Y_i - \varphi^\top(X_i)\widehat{\beta}_a(t_j)\}^2\widehat{\Sigma}_a^{-1}(t_j).$$

Under the given conditions in Theorem 1, it is to converge to

$$\frac{\sqrt{N(t_{k_1})}}{\sqrt{N(t_{k_2})}} \sum_{a=0}^{1} \Sigma_a^{-1}\Phi(a)\Sigma_a^{-1} = \text{Cov}(G(t_{k_1}), G(t_{k_2})).$$

This means $\{\sqrt{N(t_k)}(\widehat{\beta}_1^{\text{MB}*}(t_k) - \widehat{\beta}_0^{\text{MB}*}(t_k))\}_k$ and $\{G(t_k)\}_k$ have the same asymptotic distribution. Consequently, $\{\sqrt{N(t_k)}\widehat{S}^{\text{MB}*}(t_k)\}_{k=1}^{K}$ can be used to approximate the joint distribution of $\{\sup_{x \in \mathbb{X}} \varphi^\top(x)G(t_k)\}_{k=1}^{K}$.

To choose $\{z_k\}_k$ that satisfies equation 8, we adopt the $\alpha$-spending approach that allocates the total allowable type I error at each interim stage according to an error-spending function. This guarantees our test controls the type-I error. We begin by specifying an $\alpha$ spending function $\alpha(t)$ that is non-increasing and satisfies $\alpha(0) = 0$, $\alpha(T) = \alpha$. Popular choices of $\alpha(\cdot)$ include

$$\alpha_1(t) = \alpha \log\left(1 + (e-1)\frac{t}{T}\right), \qquad \alpha_2(t) = 2 - 2\Phi\left(\frac{\Phi^{-1}(1 - \alpha/2)\sqrt{T}}{\sqrt{t}}\right),$$

$$\alpha_3(t) = \alpha\left(\frac{t}{T}\right)^\theta, \quad \text{for } \theta > 0, \qquad \alpha_4(t) = \alpha\frac{1 - \exp(-\gamma t/T)}{1 - \exp(-\gamma)}, \quad \text{for } \gamma \neq 0, \tag{9}$$

where $\Phi(\cdot)$ denotes the cumulative distribution function of a standard normal variable and $\Phi^{-1}(\cdot)$ is its quantile function. Based on $\alpha(\cdot)$, we iteratively calculate $\widehat{z}_k$, $k = 1, \ldots, K$ as the solution of

$$\text{Pr}^*\left\{\max_{j \in \{1,\ldots,k-1\}}\left(\sqrt{N(t_j)}\widehat{S}^{\text{MB}*}(t_j) - \widehat{z}_j\right) \leq 0, \sqrt{N(t_k)}\widehat{S}^{\text{MB}*}(t_k) > \widehat{z}_k\right\} = \alpha(t_k) - \alpha(t_{k-1}), \tag{10}$$

and reject $H_0$ when $\sqrt{N(t_k)}S(t_k) > \widehat{z}_k$ holds for some $k$.

The validity of the bootstrap test is summarized in Theorems 3 and 4 below.

**Theorem 3** *Assume the conditions in Theorem 1 hold. Assume $q = O(N^{\alpha^*}(t_1))$ for some $0 < \alpha^* < 1/3$, almost surely. Then conditional on the counting process $N(\cdot)$, we have*

$$\sup_{z_1,\ldots,z_K}\left|\text{Pr}^*\left\{\max_{k \in \{1,\ldots,K\}}\left(\sqrt{N(t_k)}\widehat{S}^{\text{MB}*}(t_k) - z_k\right) > 0\right\} - \text{Pr}\left\{\max_{k \in \{1,\ldots,K\}}\left(\sup_{x \in \mathbb{X}}\varphi^\top(x)G(t_k) - z_k\right) > 0\right\}\right|$$

$$\leq c\left[q^{1/2}N^{-1/6}(t_1)\log^{11/6}\{KN(t_1)\} + qN^{-\alpha_0/3}(t_1)\log^{(5+\alpha_0)/3}\{KN(t_1)\}\right]$$

*for some constant $c > 0$ with probability at least $1 - O(N^{-\alpha_0}(t_1))$, where $\text{Pr}^*(\cdot)$ denotes the probability measure conditional on the data stream $\{X_i, A_i, Y_i\}_{i=1}^{+\infty}$.*

**Theorem 4** *Assume the conditions in Theorem 3 hold. Then conditional on $N(\cdot)$, the critical values $\{\widehat{z}_k\}_k$ satisfy*

$$\left|\text{Pr}\left\{\max_{k \in \{1,\ldots,K\}}\left(\sup_{x \in \mathbb{X}}\varphi^\top(x)G(t_k) - \widehat{z}_k\right) > 0\right\} - \alpha\right|$$

$$\leq c\left[q^{1/2}N^{-1/6}(t_1)\log^{11/6}\{KN(t_1)\} + qN^{-\alpha_0/3}(t_1)\log^{(5+\alpha_0)/3}\{KN(t_1)\}\right], \tag{11}$$

*for some constant $c > 0$.*

When the RHS of equation 11 is $o_p(1)$, it follows from Theorems 2 and 4 that our test is valid. The conditional distribution in equation 10 can be approximated by the empirical distribution of Bootstrap samples.

Finally, we remark that our test can be online updated as batches of observations arrive at the end of each interim stage. A pseudocode summarizing our procedure is given in Algorithm 1 in the appendix. The spatial complexity of the proposed algorithm is $O(B)$, where $B$ is the number of bootstrap samples. The time complexity is $O(Bk + N(t_k))$ up to the $k$-th interim stage. Suppose $N(t_j) - N(t_{j-1}) = n$ for any $1 \leq j \leq K$, we have $Bk + N(t_k) = (B + n)k \ll Bnk = BN(t_k)$ for large $n$ and $B$. Hence, our procedure is much faster compared to the standard wild bootstrap.

## 4 NUMERICAL STUDIES

### 4.1 SIMULATION STUDIES

In this section, we conduct Monte Carlo simulations to examine the finite sample properties of the proposed test. We generated the potential outcomes as $Y_i^*(a) = 1 + (X_{i1} - X_{i2})/2 + a\tau(X_i) + \varepsilon_i$, where $\varepsilon_i$'s are i.i.d $N(0, 0.5^2)$. The covariates $X_i = (X_{i1}, X_{i2}, X_{i3})^\top$ were generated as follows. We first generated $X_i^* = (X_{i1}^*, X_{i2}^*, X_{i3}^*)^\top$ from a multivariate normal distribution with zero mean and covariance matrix equal to $\{0.5^{|i-j|}\}_{i,j}$. Then we set $X_{ij} = X_{ij}^* \mathbb{I}(X_{ij}^*| \leq 2) + 2\text{sgn}(X_{ij}^*)\mathbb{I}(X_{ij}^*| > 2)$. We consider two randomization designs. In the first design, the treatment assignment is nondynamic and completely random. Specifically we set $\pi_i(a, x) = 0.5$, for any $a, x$ and $i$. In the second design, we use an $\epsilon$-greedy strategy to generate the treatment with $\varepsilon = 0.3$. In addition, we set $N(T_1) = 2000$ and $N(T_j) - N(T_{j-1}) = 2n$ for $2 \leq j \leq K$ and some $n > 0$. We consider two combinations of $(n, K)$, corresponding to $(n, K) = (200, 5)$ and $(20, 50)$.

We set the significance level $\alpha = 0.05$ and choose $B = 10000$. We set $\tau(X_i) = \phi_\delta\{(X_{i1} + X_{i2})/\sqrt{2}\}X_{i3}^2$ for some function $\phi_\delta$ parameterized by some $\delta \geq 0$. We consider two scenarios for $\phi_\delta$. Specifically, we set $\phi_\delta(x) = \delta x^2/3$ in Scenario 1 and $\phi_\delta = \delta \cos(\pi x)$ in Scenario 2. For each setting, we further consider four cases by setting $\delta = 0, 0.1, 0.15, 0.2, 0.25$ and $0.3$. When $\delta = 0$, $H_0$ holds. Otherwise, $H_1$ holds. For all settings, we construct the basis function $\varphi(\cdot)$ using additive cubic splines. For each univariate spline, we set the number of internal knots to be 4. These knots are equally spaced between $[-2, 2]$.

We denote our test by BAT, short for bootstrap-assisted test. We run our experiments on a single computer instance with 40 Intel(R) Xeon(R) 2.20GHz CPUs. It takes 1-2 seconds on average to compute each test. In Table 1 (see Appendix G), we report the rejection probabilities and average stopping times (defined as the average number of samples consumed when the experiment is terminated) of the proposed test aggregated over 400 simulations when $\alpha_1(\cdot)$ is chosen as the spending function. In Figure 2, we plot the rejection probabilities of our tests and the average stopping times of the experiments. It can be seen that the type-I error rates are close to the nominal level in all cases. The power of our test increases as $\delta$ increases, demonstrating its consistency. In addition, when $\delta > 0$, our experiments are stopped early in all cases.

To further evaluate our method, we compare it with a test based on the law of iterated logarithm (denoted by LIL). LIL determines the decision boundary based on an always valid finite error bound (see Appendix F for details about the competing method). It can be seen from Figure 2 that our method has much larger power than the law of iterated logarithm approach.

### 4.2 REAL DATA ANALYSIS

In this section, we apply the proposed method to a Yahoo! Today Module user click log dataset[1], which contains 45,811,883 user visits to the Today Module, during the first ten days in May 2009. For the $i$th visit, the dataset contains an ID of the new article recommended to the user, a binary response variable $Y_i$ indicating whether the user clicked the article or not, and a five dimensional feature vector summarizing information of the user. Due to privacy concerns, feature definitions and article names were not included in the data. Each feature vector sums up to 1. Therefore, we took the first three and the fifth elements to form the covariates $X_i$. For illustration, we only consider a

---

[1]https://webscope.sandbox.yahoo.com/catalog.php?datatype=r&did=49

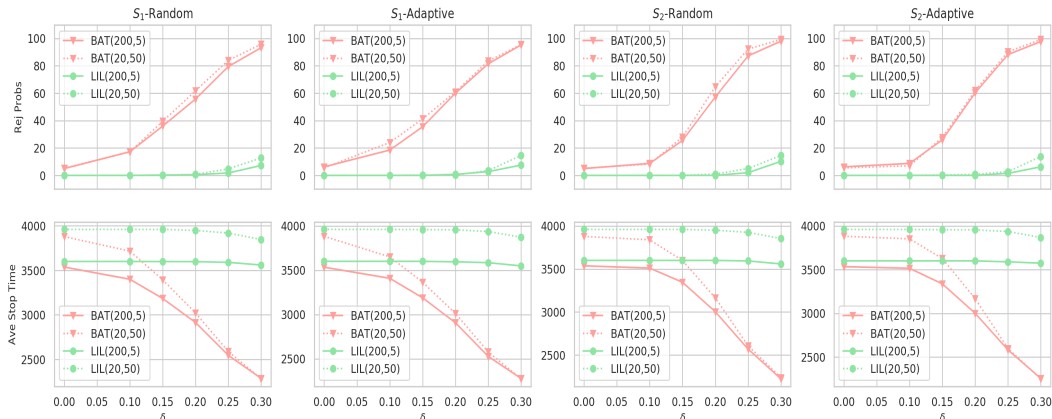

Figure 2: Rejection probabilities and average stopping times of the proposed test when $\alpha_1(\cdot)$ is chosen as the spending function. From left to right: Scenario 1 with random design, Scenario 1 with $\epsilon$-greedy design, Scenario 2 with random design and Scenario 2 with $\epsilon$-greedy design.

subset of data that contains visits on May 1st where the recommended article ID is either 109510 or 109520. These two articles were being recommended most on that day. This gives us a total of 405888 visits. On the reduced dataset, define $A_i = 1$ if the recommended article is 109510 and $A_i = 0$ otherwise.

We first conduct A/A experiments (which compare these two articles against themselves) to examine the validity of our test. The A/A experiments are done when every 2000 more users are available, we randomly assign 1000 users to arm A, and the other 1000 users in arm B. We expect our test will not reject $H_0$ in A/A experiments, since the articles being recommended are the same. Then, we conduct A/B experiment to test the QTE of these two articles. The test statistics and their corresponding critical values are plotted in Figure 3. On average it takes several seconds to implement our test. It can be seen that our test is able to be reject $H_0$ after obtaining the first one-third of the observations, in the A/B experiment. In the A/A experiments, we fail to reject $H_0$, as expected.

## 5 DISCUSSION

In this paper, we propose a new testing procedure for evaluating the performance of technology products in tech companies based on the notion of qualitative treatment effects. Currently, we only focus on comparing two products. It would be practically useful to develop a multiple testing procedure under settings with multiple treatment options. These topics warrant further investigations.

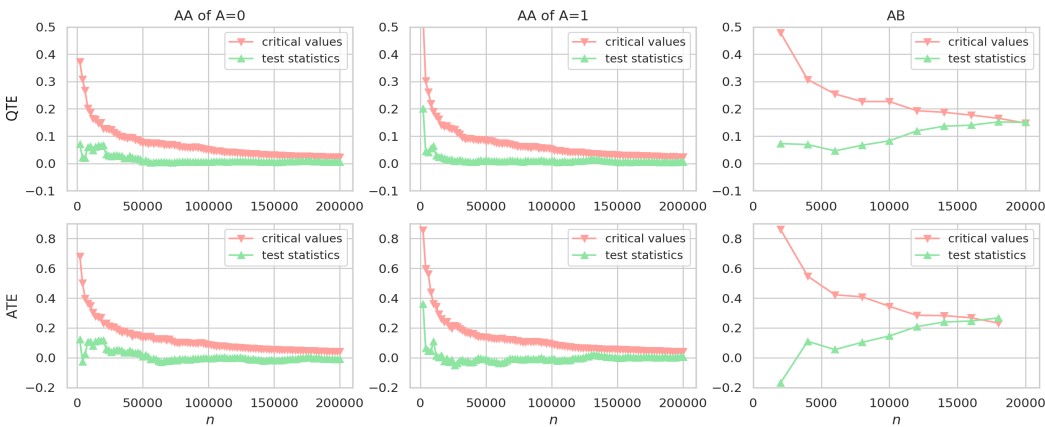

Figure 3: Critical values and test statistics.

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

**Input:** Number of bootstrap samples $B$, an $\alpha$ spending function $\alpha(\cdot)$.

**Initialize:** $n = 0, \widehat{\Sigma}_0 = \widehat{\Sigma}_1 = O_{p+1}, \widehat{\gamma}_0 = \widehat{\gamma}_1 = 0_{p+1}, \widehat{\beta}_{0,b} = \widehat{\beta}_{1,b} = 0_{p+1}$, and a set $\mathcal{I} = \{1, \ldots, B\}$.

**For** $k = 1$ to $K$ **do**

    **Initialize:** $m = 0$ and $\widehat{\Phi}_0 = \widehat{\Phi}_1 = O_{p+1}$.

    **Step 1: Online update of** $\widehat{\beta}_a$

    **For** $i = N(t_{k-1}) + 1$ to $N(t_k)$ **do**

        $n = n + 1$ and $m = m + 1$;

        $\widehat{\Sigma}_a = (1 - n^{-1})\widehat{\Sigma}_a + n^{-1}\varphi(X_i)\varphi^\top(X_i)\mathbb{I}(A_i = a), a = 0, 1$;

        $\widehat{\gamma}_a = (1 - n^{-1})\widehat{\gamma}_a + n^{-1}\varphi(X_i)Y_i\mathbb{I}(A_i = a), a = 0, 1$;

    **Compute** $\widehat{\beta}_a = \widehat{\Sigma}_a^{-1}\widehat{\gamma}_a$ for $a \in \{0, 1\}$ and $S = \sup_{x \in \mathbb{X}} \varphi^\top(x)(\widehat{\beta}_1 - \widehat{\beta}_0)$;

    **Step 2: Bootstrap**

    **For** $i = N(t_{k-1}) + 1$ to $N(t_k)$ **do**

        $\widehat{\Phi}_a = \widehat{\Phi}_a + \widehat{\Sigma}_a^{-1}\varphi(X_i)\varphi^\top(X_i)\{Y_i - \varphi^\top(X_i)\widehat{\beta}_a\}^2\widehat{\Sigma}_a^{-1}\mathbb{I}(A_i = a), a = 0, 1$;

    **For** $b = 1, \ldots, B$ **do**

        Generate two independent $N(0, I_{p+1})$ Gaussian vectors $e_0, e_1$;

        $\widehat{\beta}_{a,b} = (1 - mn^{-1})\widehat{\beta}_{a,b} + n^{-1}\widehat{\Phi}_a^{1/2}e_a, a = 0, 1$;

        **Compute** $\widehat{S}_b = \sup_{x \in \mathbb{X}} \varphi^\top(x)(\widehat{\beta}_{1,b} - \widehat{\beta}_{0,b})$;

    **Step 3: Reject or not**

    Set $z$ to be the upper $\{\alpha(t) - |\mathcal{I}|^c/B|\}/(1 - |\mathcal{I}^c|/B)$-th percentile of $\{\widehat{S}_b\}_{b \in \mathcal{I}}$;

    Update $\mathcal{I}$ as $\mathcal{I} \leftarrow \{b \in \mathcal{I} : \widehat{S}_b \le z\}$.

    **If** $S > z$: Reject $H_0$ and terminate the experiment.

    **Algorithm 1:** the Pseudocode that summarizing the online bootstrap testing procedure.

Baqun Zhang, Anastasios A Tsiatis, Eric B Laber, and Marie Davidian. Robust estimation of optimal dynamic treatment regimes for sequential treatment decisions. *Biometrika*, 100(3):681–694, 2013.

Li-xin Zhang. Strong approximations of martingale vectors and their applications in Markov-chain adaptive designs. *Acta Math. Appl. Sin. Engl. Ser.*, 20(2):337–352, 2004. ISSN 0168-9673. doi: 10.1007/s10255-004-0173-z. URL https://doi.org/10.1007/s10255-004-0173-z.

## A  NOTATIONS

We introduce some general notations used in the appendix. For any matrix Mat, we use $\|\text{Mat}\|_p$ to denote the matrix norm induced by the corresponding $\ell_p$ norm of vectors, for $1 \le p \le +\infty$. For two nonnegative sequences $\{s_{1,n}\}_n$ and $\{s_{2,n}\}_n$, we will use the notation $s_{1,n} \preceq s_{2,n}$ to represent that $s_{1,n} \le \bar{c}s_{2,n}$ for some universal constant $\bar{c} > 0$ whose value is allowed to change from place to place. When a matrix Mat is degenerate, $\text{Mat}^{-1}$ denotes the Moore-Penrose inverse of Mat. For any vector $\psi$, we use $\psi^{(i)}$ to denote its $i$-th element.

In Algorithm 1, we use $O_{p+1}$ to denote a $(p + 1) \times (p + 1)$ zero matrix and $0_{p+1}$ to denote a $(p + 1)$-dimensional zero vector.

## B  MORE ON THE BASIS FUNCTION

### B.1  CONDITION (A3)

(A3). Assume $\lambda_{\min}[\mathrm{E}\varphi(X)\varphi^\top(X)] \asymp 1$, $\lambda_{\max}[\mathrm{E}\varphi(X)\varphi^\top(X)] \asymp 1$, $\sup_x \|\varphi(x)\|_1 = O(q^{1/2})$, $\liminf_q \inf_{x \in \mathbb{X}} \|\varphi(x)\|_2 > 0$. In addition, assume

$$\sup_{\substack{x,y \in \mathbb{X} \\ x \ne y}} \frac{\|\varphi(x) - \varphi(y)\|_2}{\|x - y\|_2} \preceq q^{1/2}. \tag{12}$$

When a tensor-product B-spline is used (see Section 6 of Chen & Christensen, 2015, for a brief overview of tenor-product B-splines), (A3) is automatically satisfied. Specifically, $\lambda_{\min}[\mathrm{E}\varphi(X)\varphi^\top(X)] \asymp 1$, $\lambda_{\max}[\mathrm{E}\varphi(X)\varphi^\top(X)] \asymp 1$ follow from Theorem 3.3 of (Burman & Chen, 1989). $\sup_x \|\varphi(x)\|_1 = O(q^{1/2})$ follows by noting that the absolute value of each element in $\varphi(x)$ is bounded by some universal constant. $\liminf_q \inf_{x\in\mathbb{X}} \|\varphi(x)\|_2 > 0$ follows from the arguments used in the proof of Lemma E.4 of Shi et al. (2020b). The last condition in equation 12 holds by noting that each function in the vector $\varphi(\cdot)$ is Lipschitz continuous when a tensor-product B-spline is used.

## B.2 On the approximation error

The proposed test remains valid as long as the approximation error satisfies

$$\inf_{\beta_0,\beta_1\in\mathbb{R}^p} \sup_{x\in\mathbb{X},a\in\{0,1\}} |Q_0(x,a) - Q(x,a;\beta_0,\beta_1)| = o(\{N(T)\}^{-1/2}), \tag{13}$$

with probability tending to 1. In the following, we introduce some sufficient conditions for equation 13.

Suppose the Q-function $Q_0(\cdot,a)$ is $p$-smooth (see the definition of $p$-smoothness in Stone, 1982), for $a \in [0,1]$. When a tensor-product B-spline or Wavelet basis is used for $\varphi(\cdot)$, then there exist some $\beta_0^*$ and $\beta_1^*$ that satisfy

$$\inf_{\beta_0,\beta_1\in\mathbb{R}^p} \sup_{x\in\mathbb{X},a\in\{0,1\}} |Q_0(x,a) - Q(x,a;\beta_0,\beta_1)| = O(q^{-p/d}).$$

See Section 2.2 of Huang (1998) for detailed discussions on the approximation power of these basis functions. Condition equation 13 is thus automatically satisfied when

$$q \gg \{N(T)\}^{d/(2p)},$$

with probability tending to 1.

## C Adaptive randomization

In practice, the company might want to allocate more traffic to a better treatment based on the observed data stream. The $\epsilon$-greedy strategy is commonly used to balance the trade-off between exploration and exploitation. For a given $0 < \varepsilon_0 < 1$, consider the following randomization procedure: for some integer $N_0 > 0$ and any $j \geq N_0$, $a \in \{0,1\}$, $x \in \mathbb{X}$, we set

$$\pi_{j-1}(a,x) = (1-\varepsilon_0)a\mathbb{I}\{\varphi^\top(x)(\widehat{\beta}_{1,j-1} - \widehat{\beta}_{0,j-1}) > 0\} + \varepsilon_0(1-a)\mathbb{I}\{\varphi^\top(x)(\widehat{\beta}_{1,j-1} - \widehat{\beta}_{0,j-1}) \leq 0\},$$

where

$$\widehat{\beta}_{a,j} = \widehat{\Sigma}_{a,j}^{-1}\frac{1}{j}\sum_{i=1}^{j}\{\mathbb{I}(A_i = a)\varphi(X_i)Y_i\} \quad \text{and} \quad \widehat{\Sigma}_{a,j} = \frac{1}{j}\sum_{i=1}^{j}\mathbb{I}(A_i = a)\varphi(X_i)\varphi^\top(X_i).$$

It is immediate to see that $\widehat{\Sigma}_a(t) = \widehat{\Sigma}_{a,n(t)}$ and $\widehat{\beta}_a(t) = \widehat{\beta}_{a,n(t)}$. Define

$$\pi^*(a,x) = (1-\varepsilon_0)a\mathbb{I}\{\varphi^\top(x)(\beta_1 - \beta_0) > 0\} + \varepsilon_0(1-a)\mathbb{I}\{\varphi^\top(x)(\beta_1 - \beta_0) \leq 0\}$$

for any $a \in \{0,1\}$ and $x \in \mathbb{X}$.

**Lemma 2** *Assume (A1)-(A3) hold. Assume $\inf_{x\in\mathbb{X}} \pi^*(a,x) > 0$ and $|Y^*(a)|$ is bounded almost surely, for $a \in \{0,1\}$. Assume $Pr(|\varphi^\top(X)(\beta_1 - \beta_0)| \leq \epsilon) \leq L_0\epsilon$, for some constant $L_0 > 0$ and any $\epsilon > 0$. Then for any $\{j_n\}_n$ that satisfies $\sqrt{j_n}/\sqrt{\log j_n} \gg q^2$, the following event occurs with probability at least $1 - O(j_n^{-1})$,*

$$\sum_{a\in\{0,1\}} E^{\mathcal{F}_{i-1}}\left|\sum_{i=1}^{k}\{\pi_{i-1}(a,X) - \pi^*(a,X)\}\right| \preceq q\sqrt{k\log k}, \quad \forall k \geq j_n.$$

Lemma 2 implies that Condition equation 5 in Theorem 1 automatically holds with $\alpha_0 = 1/2$, when the epsilon-greedy strategy is used. When $\varphi^\top(X)(\beta_1 - \beta_0)$ is a continuous random variable, the assumption $Pr(|\varphi^\top(X)(\beta_1 - \beta_0)| \leq \epsilon) \leq L_0\epsilon$ for any $\epsilon > 0$ in Lemma 2 is satisfied if $\varphi^\top(X)(\beta_1 - \beta_0)$ has a bounded probability density function. When $\varphi^\top(X)(\beta_1 - \beta_0)$ is discrete, this assumption is satisfied if $\inf_{x\in\mathbb{X}} |\varphi^\top(x)(\beta_1 - \beta_0)| > 0$.

# D  TESTING THE AVERAGE TREATMENT EFFECTS

## D.1  THE ALGORITHM

We focus on testing the following hypothesis,

$$H_0 : \mathrm{E}Y_i^*(1) \leq \mathrm{E}Y_i^*(0) \quad \text{versus} \quad H_1 : \mathrm{E}Y_i^*(1) > \mathrm{E}Y_i^*(0).$$

Under (A1) and (A2), it suffices to test

$$H_0 : \mathrm{E}Q(X_i, 1) \leq \mathrm{E}Q(X_i, 0) \quad \text{versus} \quad H_1 : \mathrm{E}Q(X_i, 1) > \mathrm{E}Q(X_i, 0).$$

We similarly use basis approximations to model the Q-function. Our proposal is summarized in the following algorithm. We next conduct simulation studies to evaluate this algorithm.

---

**Input:** Number of bootstrap samples $B$, an $\alpha$ spending function $\alpha(\cdot)$.
**Initialize:** $n = 0$, $\widehat{\Sigma}_0 = \widehat{\Sigma}_1 = O_{p+1}$, $\widehat{\gamma}_0 = \widehat{\gamma}_1 = 0_{p+1}$, $\widehat{\beta}_{0,b} = \widehat{\beta}_{1,b} = 0_{p+1}$, $\bar{\varphi} = 0$ and a set $\mathcal{I} = \{1, \ldots, B\}$.
**For** $k = 1$ to $K$ **do**
  **Initialize:** $m = 0$, $\widehat{\phi} = 0$ and $\widehat{\Phi}_0 = \widehat{\Phi}_1 = O_{p+1}$.

  **For** $i = N(t_{k-1}) + 1$ to $N(t_k)$ **do**
    $n = n + 1$, $m = m + 1$ and $\bar{\varphi} = n^{-1}(n - 1)\bar{\varphi} + n^{-1}\varphi(X_i)$;
    $\widehat{\Sigma}_a = (1 - n^{-1})\widehat{\Sigma}_a + n^{-1}\varphi(X_i)\varphi^\top(X_i)\mathbb{I}(A_i = a), a = 0, 1$;
    $\widehat{\gamma}_a = (1 - n^{-1})\widehat{\gamma}_a + n^{-1}\varphi(X_i)Y_i\mathbb{I}(A_i = a), a = 0, 1$;
  **Compute** $\widehat{\beta}_a = \widehat{\Sigma}_a^{-1}\widehat{\gamma}_a$ for $a \in \{0, 1\}$ and $S = \bar{\varphi}^\top(\widehat{\beta}_1 - \widehat{\beta}_0)$;

  **For** $i = N(t_{k-1}) + 1$ to $N(t_k)$ **do**
    $\widehat{\phi} = \widehat{\phi} + [\{\varphi(X_i) - \bar{\varphi}\}^\top(\widehat{\beta}_1 - \widehat{\beta}_0)]^2$.
    $\widehat{\Phi}_a = \widehat{\Phi}_a + \widehat{\Sigma}_a^{-1}\varphi(X_i)\varphi^\top(X_i)\{Y_i - \varphi^\top(X_i)\widehat{\beta}_a\}^2\widehat{\Sigma}_a^{-1}\mathbb{I}(A_i = a), a = 0, 1$;
  **For** $b = 1, \ldots, B$ **do**
    Generate two independent $N(0, I_{p+1})$ Gaussian vectors $e_0, e_1$, $N(0, 1)$ random variable $e_2$;
    $\widehat{\beta}_{a,b} = (1 - mn^{-1})\widehat{\beta}_{a,b} + n^{-1}\widehat{\Phi}_a^{1/2}e_a + n^{-1}\widehat{\phi}^{1/2}e_2, a = 0, 1$;
    **Compute** $\widehat{S}_b = \bar{\varphi}^\top(\widehat{\beta}_{1,b} - \widehat{\beta}_{0,b})$;

  Set $z$ to be the upper $\{\alpha(t) - |\mathcal{I}|^c/B|\}/(1 - |\mathcal{I}^c|/B)$-th percentile of $\{\widehat{S}_b\}_{b\in\mathcal{I}}$;
  Update $\mathcal{I}$ as $\mathcal{I} \leftarrow \{b \in \mathcal{I} : \widehat{S}_b \leq z\}$.
  **If** $S > z$:
    Reject $H_0$ and terminate the experiment;

---

## D.2  NUMERICAL STUDIES

In this section, we compare our procedure with the always valid test for testing ATE (Johari et al., 2015). We generate the potential outcomes with the same model, except that $\varepsilon_i$'s are i.i.d $N(0, 1)$. However, we set $N(T_1) = 1000$ and $N(T_j) - N(T_{j-1}) = 2n$ for $2 \leq j \leq K$ and some $n > 0$. We consider two combinations of $(n, K)$, corresponding to $(n, K) = (100, 5)$ and $(10, 50)$. For all settings, we use a linear function to approximate $Q$.

In Table 2 (see Appendix G) and Figure 4, we show the rejection probabilities and average stopping times of the proposed test aggregated over 400 simulations, when $\alpha_1(\cdot)$ is chosen as the spending function. It can be seen that our method behaves better than the always valid test when the effect size is small, and comparable when the effect size is large. The always valid test fails in the adaptive randomization settings, as the type-I error rates are around 50% under the null hypothesis.

# E  PROOFS

Note that we require $\mathbb{X}$ to be a compact set. To simplify the proof, we assume $\mathbb{X} = [0, 1]^d$.

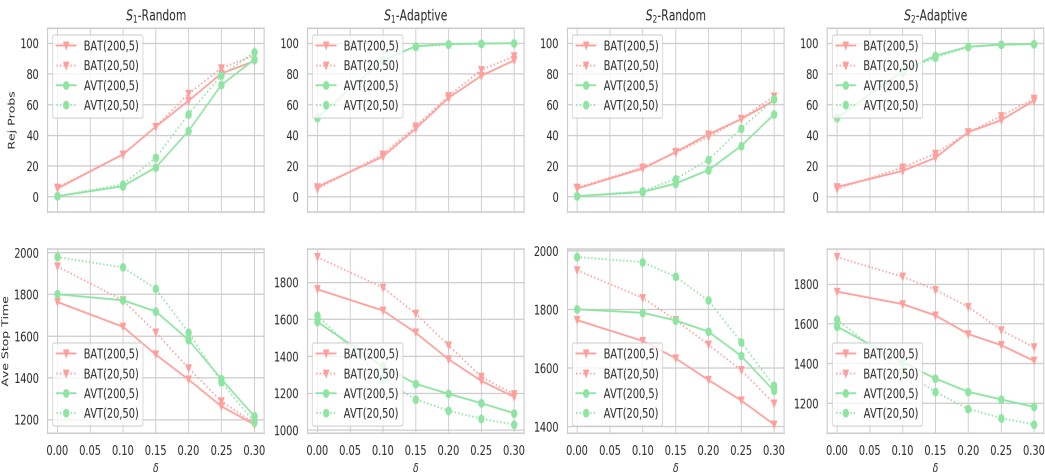

Figure 4: Rejection probabilities and average stopping times of the proposed test when $\alpha_1(\cdot)$ is chosen as the spending function.

### E.1  PROOF OF LEMMA 1

Set $\mathcal{F}_0 = \emptyset$. We state the following lemma before proving Lemma 1.

**Lemma 3** *For any* $j \geq 1$, $(X_j, Y_j^*(0), Y_j^*(1)) \perp\!\!\!\perp \mathcal{F}_{j-1}$.

For any $a \in \{0, 1\}$, $i \geq 1$, notice that

$$
\begin{aligned}
E\mathbb{I}(A_i = a)\{Y_i - \varphi^\top(X_i)\beta_a\} &= E\mathbb{I}(A_i = a)\{Y_i^*(a) - \varphi^\top(X_i)\beta_a\} \\
&= EE^{X_i, \mathcal{F}_{i-1}}[\mathbb{I}(A_i = a)\{Y_i^*(a) - \varphi^\top(X_i)\beta_a\}],
\end{aligned}
$$

where the first equation is due to Assumption (A1) and $E^{X_i, \mathcal{F}_{i-1}}$ denotes the conditional expectation given $\mathcal{F}_{i-1}$ and $X_i$. By Assumption (A2), we have

$$
E^{X_i, \mathcal{F}_{i-1}}[\mathbb{I}(A_i = a)\{Y_i^*(a) - \varphi^\top(X_i)\beta_a\}] = \{E^{X_i, \mathcal{F}_{i-1}}\mathbb{I}(A_i = a)\}[E^{X_i, \mathcal{F}_{i-1}}\{Y_i^*(a) - \varphi^\top(X_i)\beta_a\}].
$$

The second term on the RHS equals zero due to Lemma 3 and our model assumption $E\{Y_i^*(a)|X_i\} = \varphi^\top(X_i)\beta_a$. The proof is hence completed.

### E.2  PROOF OF THEOREM 1

Let $n(\cdot)$ be the realization of the counting process $N(\cdot)$. We will show the assertion in Theorem 1 holds for any such realizations that satisfy $n(t_1) < n(t_2) < \cdots < n(t_K)$. The case where some of the $n(t_k)$'s are the same can be similarly discussed.

For any $j \geq 1$, define $\sigma(\mathcal{F}_j)$ to be the $\sigma$-algebra generated by $\mathcal{F}_j$. For $a \in \{0, 1\}$, define

$$
\widehat{\Sigma}_{a,j} = \frac{1}{j}\sum_{i=1}^{j}\mathbb{I}(A_i = a)\varphi(X_i)\varphi^\top(X_i) \quad \text{and} \quad \widehat{\beta}_{a,j} = \widehat{\Sigma}_{a,j}^{-1}\left(\frac{1}{j}\sum_{i=1}^{j}\mathbb{I}(A_i = a)\varphi(X_i)Y_i\right).
$$

It is immediate to see that $\widehat{\Sigma}_a(t) = \widehat{\Sigma}_{a,n(t)}$ and $\widehat{\beta}_a(t) = \widehat{\beta}_{a,n(t)}$. Define $\delta_n = qn^{-\alpha_0}\log^{\alpha_0} n$. We state the following lemmas before proving Theorem 1.

**Lemma 4** *There exists some constant* $0 < \epsilon_0 < 1$ *such that* $\lambda_{\min}[E\varphi(X)\varphi^\top(X)] \geq \epsilon_0$, $\lambda_{\max}[E\varphi(X)\varphi^\top(X)] \leq \epsilon_0^{-1}$, $\sup_x \|\varphi(x)\|_2 \leq \sup_x \|\varphi(x)\|_1 \leq \epsilon_0^{-1}\sqrt{q}$, $\min_{a \in \{0,1\}} \lambda_{\min}[\Sigma_a] \geq \epsilon_0$, $\max_{a \in \{0,1\}} \|\beta_a\|_2 \leq \epsilon_0^{-1}$, $\max_{a \in \{0,1\}} |Y^*(a)| \leq \epsilon_0^{-1}$ *and* $\sup_x \max_{a \in \{0,1\}} |\varphi^\top(x)\beta_a| \leq \epsilon_0^{-1}$.

**Lemma 5** *Assume the conditions in Theorem 1 hold. Then for any sequence* $\{j_n\}_n$ *that satisfies* $j_n^{\alpha_0}/\log^{\alpha_0}(j_n) \gg q^2$, *we have with probability at least* $1 - O(j_n^{-\alpha_0})$ *that for any* $a \in \{0, 1\}$ *and*

*any $k \geq j_n$,*

$$\|(\widehat{\Sigma}_{a,k} - \Sigma_a)\|_2 \preceq q\delta_k + \sqrt{qk^{-1}\log k}, \tag{14}$$

$$\|(\widehat{\Sigma}_{a,k}^{-1} - \Sigma_a^{-1})\|_2 \preceq q\delta_k + \sqrt{qk^{-1}\log k}. \tag{15}$$

**Lemma 6** *Assume the conditions in Theorem 1 hold. The for any sequence $\{j_n\}_n$ that satisfies $j_n/\log(j_n) \gg q$, we have with probability at least $1 - O(j_n^{-1})$ that for any $a \in \{0,1\}$ and any $k \geq j_n$,*

$$\left\|\sum_{i=1}^k \varphi(X_i)\mathbb{I}(A_i = a)\{Y_i - \varphi^\top(X_i)\beta_a\}\right\|_2 \preceq \sqrt{qk\log k}.$$

For $a \in \{0,1\}$,

$$\widehat{\beta}_{a,k} - \beta_a = \widehat{\Sigma}_a^{-1}\left[\frac{1}{k}\sum_{i=1}^k \mathbb{I}(A_i = a)\varphi(X_i)\{Y_i - \varphi^\top(X_i)\beta_a\}\right],$$

and hence

$$\left\|\widehat{\beta}_{a,k} - \beta_a - \Sigma_a^{-1}\left[\frac{1}{k}\sum_{i=1}^k \mathbb{I}(A_i = a)\varphi(X_i)\{Y_i - \varphi^\top(X_i)\beta_a\}\right]\right\|_2 \tag{16}$$

$$\leq \|\widehat{\Sigma}_{a,k}^{-1} - \Sigma_a^{-1}\|_2 \left\|\frac{1}{k}\sum_{i=1}^k \mathbb{I}(A_i = a)\varphi(X_i)\{Y_i - \varphi^\top(X_i)\beta_a\}\right\|_2$$

$$\preceq (q\delta_k + \sqrt{qk^{-1}\log k})q^{1/2}k^{-1/2}\log^{1/2} k, \quad \forall k \geq j_n,$$

with probability at least $1 - O(j_n^{-\alpha_0})$, by Lemma 5 and Lemma 6. Define

$$B^*(t) = \frac{1}{\sqrt{n(t)}}\sum_{i=1}^{n(t)}[\Sigma_1^{-1}\varphi(X_i)A_i\{Y_i - \varphi^\top(X_i)\beta_1\} - \Sigma_0^{-1}\varphi(X_i)(1 - A_i)\{Y_i - \varphi^\top(X_i)\beta_0\}].$$

It follows that

$$\|B^*(t_k) - B(t_k)\|_2 \preceq \{q^{3/2}\delta_{n(t_k)} + q\sqrt{n^{-1}(t_k)\log n(t_k)}\}n^{-1/2}(t_k)\log^{1/2} n(t_k), \forall k \geq 1, \tag{17}$$

with probability at least $1 - O(n^{-\alpha_0}(t_1))$, and hence

$$\|\sup_{x\in\mathbb{X}}\varphi^\top(x)B^*(t_k) - \sup_{x\in\mathbb{X}}\varphi^\top(x)B(t_k)\|_2 \leq \bar{c}\{q^2\delta_{n(t_k)} + q^{3/2}\sqrt{n^{-1}(t_k)\log n(t_k)}\}\sqrt{n^{-1}(t_k)\log n(t_k)},$$
$$\forall k \geq 1,$$

with probability at least $1 - O(n^{-\alpha_0}(t_1))$, for some constant $\bar{c} > 0$, by equation 41. Under the given conditions on $q$ and $n(t_1)$, we have

$$q\sqrt{n^{-1}(t_k)\log n(t_k)} = o(1), \quad \forall k \geq 1,$$

and hence

$$\|\sup_{x\in\mathbb{X}}\varphi^\top(x)B^*(t_k) - \sup_{x\in\mathbb{X}}\varphi^\top(x)B(t_k)\|_2 \leq \bar{c}\{q\delta_{n(t_k)} + \sqrt{qn^{-1}(t_k)\log n(t_k)}\}, \quad \forall k \geq 1,$$

Thus, for any given $z_1, z_2, \ldots, z_K$, we obtain

$$\Pr\left\{\max_{k\in\{1,\ldots,K\}}\left(\sup_{x\in\mathbb{X}}\varphi^\top(x)B^*(t_k) - z_{k,-}\right) \leq 0\right\} - O(n^{-\alpha_0}(t_1))$$

$$\leq \Pr\left\{\max_{k\in\{1,\ldots,K\}}\left(\sup_{x\in\mathbb{X}}\varphi^\top(x)B(t_k) - z_k\right) \leq 0\right\} \tag{18}$$

$$\leq \Pr\left\{\max_{k\in\{1,\ldots,K\}}\left(\sup_{x\in\mathbb{X}}\varphi^\top(x)B^*(t_k) - z_{k,+}\right) \leq 0\right\} + O(n^{-\alpha_0}(t_1)),$$

where

$$z_{k,-} = z_k - \bar{c}\{q\delta_{n(t_k)} + \sqrt{qn^{-1}(t_k)\log n(t_k)}\},$$
$$z_{k,+} = z_k + \bar{c}\{q\delta_{n(t_k)} + \sqrt{qn^{-1}(t_k)\log n(t_k)}\}.$$

For any $i \geq 1$, $1 \leq k \leq K$, define a $q$-dimensional vector

$$\xi_{i,k} = \frac{1}{\sqrt{n(t_k)}}[\Sigma_1^{-1}\varphi(X_i)A_i\{Y_i - \varphi^\top(X_i)\beta_1\} - \Sigma_0^{-1}\varphi(X_i)(1 - A_i)\{Y_i - \varphi^\top(X_i)\beta_0\}]\mathbb{I}(i \leq n(t_k)),$$

or equivalently,

$$\xi_{i,k} = \frac{1}{\sqrt{n(t_k)}}[\Sigma_1^{-1}\varphi(X_i)A_i\{Y_i^*(1) - \varphi^\top(X_i)\beta_1\} - \Sigma_0^{-1}\varphi(X_i)(1 - A_i)\{Y_i^*(0) - \varphi^\top(X_i)\beta_0\}]\mathbb{I}(i \leq n(t_k)),$$

by Condition (A1). Let $\boldsymbol{\xi}_i = (\xi_{i,1}^\top, \xi_{i,2}^\top, \cdots, \xi_{i,K}^\top)^\top$ and $\mathcal{M}_j = \sum_{i=1}^j \boldsymbol{\xi}_i$. The sequence $\{\mathcal{M}_i\}_{i\geq 1}$ forms a multivariate martingale with respect to the filtration $\{\sigma(\mathcal{F}_i) : i \geq 1\}$, since

$$\mathrm{E}(\xi_{i,k}|\mathcal{F}_i) = [\{\mathrm{E}(\xi_{i,k}|A_i, X_i, \mathcal{F}_i)\}|\mathcal{F}_i] = 0,$$

by (A2). Let $n(t_0) = 0$. For any $i$ such that $n(t_{k-1}) < i \leq n(t_k)$ for some $1 \leq k \leq K$, we have

$$\|\boldsymbol{\xi}_i\|_\infty \leq \frac{1}{\sqrt{n(t_k)}}\{\|\Sigma_1^{-1}\varphi(X_i)\{Y_i^*(1) - \varphi^\top(X_i)\beta_1\}\|_2 + \|\Sigma_0^{-1}\varphi(X_i)\{Y_i^*(0) - \varphi^\top(X_i)\beta_0\}\|_2\}$$

$$\leq 4\sqrt{q}n^{-1/2}(t_k)\epsilon_0^{-3},$$

where the second inequality is due to Lemma 4. Therefore,

$$\mathrm{E}\|\boldsymbol{\xi}_i\|_\infty^3 \preceq \frac{q^{3/2}}{n^{3/2}(t_k)}.$$

It follows that

$$\sum_{i=1}^{n(t_K)} \mathrm{E}\|\boldsymbol{\xi}_i\|_\infty^3 = \sum_{k=1}^K \sum_{i=n(t_{k-1})+1}^{n(t_k)} \mathrm{E}\|\boldsymbol{\xi}_i\|_\infty^3 \preceq q^{3/2} \sum_{k=1}^K \frac{n(t_k) - n(t_{k-1})}{n^{3/2}(t_k)} \tag{19}$$

$$\leq \frac{q^{3/2}}{\sqrt{n(t_1)}} + q^{3/2}\sum_{k=2}^K \frac{n(t_k) - n(t_{k-1})}{n^{3/2}(t_k)} \leq q^{3/2}n^{-1/2}(t_1) + q^{3/2}\int_{n(t_1)}^{+\infty} x^{-3/2}dx = 3q^{3/2}n^{-1/2}(t_1).$$

Define a sequence of independent Gaussian vectors $\{\boldsymbol{\eta}_i\}_{i\geq 1}$ that satisfy $\boldsymbol{\eta}_i \sim N(0, \mathrm{E}(\boldsymbol{\xi}_i\boldsymbol{\xi}_i^\top|\mathcal{F}_{i-1}))$ for any $i \geq 1$. Then the distribution of $\boldsymbol{\eta}_i$ is the same as

$$\left(\frac{\mathbb{I}(i \leq n(t_1))}{\sqrt{n(t_1)}}Z^\top, \frac{\mathbb{I}(i \leq n(t_2))}{\sqrt{n(t_2)}}Z^\top, \cdots, \frac{\mathbb{I}(i \leq n(t_K))}{\sqrt{n(t_K)}}Z^\top\right),$$

where $Z$ is a $p$-dimensional mean-zero Gaussian vector with covariance matrix

$$\mathrm{Cov}[\sum_{a\in\{0,1\}} \Sigma_a^{-1}\varphi(X_i)\mathbb{I}(A_i = a)\{Y_i^*(a) - \varphi^\top(X_i)\beta_a\}|\mathcal{F}_{i-1}] \tag{20}$$

$$= \sum_{a\in\{0,1\}} \Sigma_a^{-1}\mathrm{E}[\varphi(X_i)\varphi^\top(X_i)\mathbb{I}(A_i = a)\{Y_i^*(a) - \varphi^\top(X_i)\beta_a\}^2|\mathcal{F}_{i-1}]\Sigma_a^{-1}$$

$$= \sum_{a\in\{0,1\}} \Sigma_a^{-1}\mathrm{E}\{\varphi(X_i)\varphi^\top(X_i)\mathbb{I}(A_i = a)\sigma^2(a, X_i)|\mathcal{F}_{i-1}\}\Sigma_a^{-1}$$

$$= \sum_{a\in\{0,1\}} \Sigma_a^{-1}\mathrm{E}\{\varphi(X_i)\varphi^\top(X_i)\pi_{i-1}(a, X_i)\sigma^2(a, X_i)|\mathcal{F}_{i-1}\}\Sigma_a^{-1}$$

$$\equiv \sum_{a\in\{0,1\}} \Sigma_a^{-1}\mathrm{E}^{\mathcal{F}_{i-1}}\pi_{i-1}(a, X)\sigma^2(a, X)\varphi(X)\varphi^\top(X)\Sigma_a^{-1},$$

where the second equality follows from (A2) and Lemma 3, the third equality is due to the definition of $\pi_{i-1}$ and the last equality follows from Lemma 3.

Similar to equation 19, we can show that

$$\sum_{i=1}^{n(t_K)} \mathrm{E}\|\boldsymbol{\eta}_i\|_\infty^3 \preceq q^{3/2} n^{-1/2}(t_1). \tag{21}$$

Using similar arguments in equation 20, we can show that for any $1 \le k_1 \le k_2 \le K$,

$$\sum_{i=1}^{n(t_K)} \mathrm{E}\{\xi_{i,k_1}\xi_{i,k_2}^\top|\mathcal{F}_{i-1}\} = \frac{1}{\sqrt{n(t_{k_1})n(t_{k_2})}} \sum_{i=1}^{n(t_{k_1})} \sum_{a\in\{0,1\}} \Sigma_a^{-1}\mathrm{E}^{\mathcal{F}_{i-1}}\pi_{i-1}(a,X)\sigma^2(a,X)\varphi(X)\varphi^\top(X)\Sigma_a^{-1}.$$

Let

$$V(k_1, k_2) = \frac{1}{\sqrt{n(t_{k_1})n(t_{k_2})}} \sum_{i=1}^{n(t_{k_1})} \sum_{a\in\{0,1\}} \Sigma_a^{-1}\mathrm{E}^{\mathcal{F}_{i-1}}\pi^*(a,X)\sigma^2(a,X)\varphi(X)\varphi^\top(X)\Sigma_a^{-1}$$

$$= \frac{1}{\sqrt{n(t_{k_1})n(t_{k_2})}} \sum_{i=1}^{n(t_{k_1})} \sum_{a\in\{0,1\}} \Sigma_a^{-1}\Phi_a\Sigma_a^{-1} = \frac{\sqrt{n(t_{k_1})}}{\sqrt{n(t_{k_2})}} \sum_{a\in\{0,1\}} \Sigma_a^{-1}\Phi_a\Sigma_a^{-1}.$$

Consider an arbitrary sequence of $\mathbb{R}^{p+1}$ vectors $\{b_k\}_{1\le k\le K}$. Under the given conditions, we have

$$\left| b_{k_1}^\top \left( \sum_{i=1}^{n(t_K)} \mathrm{E}(\xi_{i,k_1}\xi_{1,k_2}^\top|\mathcal{F}_{i-1}) - V(k_1,k_2) \right) b_{k_2} \right|$$

$$\preceq \frac{1}{n(t_{k_1})} \sum_{a\in\{0,1\}} \left\| \sum_{i=1}^{n(t_{k_1})} \mathrm{E}^{\mathcal{F}_{i-1}}\{\pi_{i-1}(a,X) - \pi^*(a,X)\}\sigma^2(a,X)\varphi(X)\varphi^\top(X) \right\|_2 \|b_{k_1}\|_2\|b_{k_2}\|_2.$$

Define a matrix $\boldsymbol{V}$ as

$$\boldsymbol{V} = \begin{pmatrix} V(1,1) & V(1,2) & \dots & V(1,K) \\ V(2,1) & V(2,2) & \dots & V(2,K) \\ \vdots & \vdots & & \vdots \\ V(K,1) & V(K,2) & \dots & V(K,K) \end{pmatrix}. \tag{22}$$

It follows that

$$\left\| \sum_{i=1}^{n(t_K)} \mathrm{E}(\boldsymbol{\xi}_i\boldsymbol{\xi}_i^\top|\mathcal{F}_{i-1}) - \boldsymbol{V} \right\|_2 \preceq \sup_{\substack{a\in\{0,1\}\\j\ge n(t_1)}} \left\| \frac{1}{j}\sum_{i=1}^{j} \mathrm{E}^{\mathcal{F}_{i-1}}\{\pi_{i-1}(a,X) - \pi^*(a,X)\}\sigma^2(a,X)\varphi(X)\varphi^\top(X) \right\|_2.$$

Using similar arguments in proving equation 14, we can show the RHS of the above equation is upper bounded by

$$\epsilon_0^{-2}q \sup_{\substack{a\in\{0,1\}\\x\in\mathbb{X},j\ge n(t_1)}} \left| \frac{1}{j}\sum_{i=1}^{j}\{\pi_{i-1}(a,x) - \pi^*(a,x)\} \right|,$$

and hence by $\epsilon_0^{-2}q\delta_{n(t_1)}$, with probability at least $1 - O(n^{-\alpha_0}(t_1))$. Therefore, we have

$$\lambda_{\min}\left[ \boldsymbol{V} + \delta_{n(t_1)}\boldsymbol{I}_{Kp\times Kp} - \sum_{i=1}^{n(t_K)} \mathrm{E}(\boldsymbol{\xi}_i\boldsymbol{\xi}_i^\top|\mathcal{F}_{i-1}) \right] \ge 0, \tag{23}$$

with probability at least $1 - O(n^{-\alpha_0}(t_1))$, where $\boldsymbol{I}_{Kp\times Kp}$ denotes a $Kp \times Kp$ identity matrix.

Moreover, notice that

$$\sup_{\substack{a \in \{0,1\} \\ x \in \mathbb{X}, j \geq n(t_1)}} \left| \frac{1}{j} \sum_{i=1}^{j} \{\pi_{i-1}(a,x) - \pi^*(a,x)\} \right|$$

is bounded between $0$ and $1$. For any $a \in \{0,1\}$ and any $z > 0$, we have

$$\mathrm{E} \sup_{\substack{a \in \{0,1\} \\ x \in \mathbb{X}, j \geq n(t_1)}} \left| \frac{1}{j} \sum_{i=1}^{j} \{\pi_{i-1}(a,x) - \pi^*(a,x)\} \right|$$

$$\leq \mathrm{E} \sup_{\substack{a \in \{0,1\} \\ x \in \mathbb{X}, j \geq n(t_1)}} \left| \frac{1}{j} \sum_{i=1}^{j} \{\pi_{i-1}(a,x) - \pi^*(a,x)\} \right| \mathbb{I} \left( \sup_{\substack{a \in \{0,1\} \\ x \in \mathbb{X}, j \geq n(t_1)}} \left| \frac{1}{j} \sum_{i=1}^{j} \{\pi_{i-1}(a,x) - \pi^*(a,x)\} \right| \leq z \right)$$

$$+ \mathrm{Pr} \left( \sup_{\substack{a \in \{0,1\} \\ x \in \mathbb{X}, j \geq n(t_1)}} \left| \frac{1}{j} \sum_{i=1}^{j} \{\pi_{i-1}(a,x) - \pi^*(a,x)\} \right| > z \right).$$

Under the given conditions, we have

$$\mathrm{E} \sup_{\substack{a \in \{0,1\} \\ x \in \mathbb{X}, j \geq n(t_1)}} \left| \frac{1}{j} \sum_{i=1}^{j} \{\pi_{i-1}(a,x) - \pi^*(a,x)\} \right| \preceq \delta_{n(t_1)} + O(n^{-\alpha_0}(t_1)).$$

Therefore, we obtain

$$\mathrm{E} \left\| \sum_{i=1}^{n(t_K)} \mathrm{E}(\boldsymbol{\xi}_i \boldsymbol{\xi}_i^\top | \mathcal{F}_{i-1}) - \boldsymbol{V} \right\|_2 \preceq q n^{-\alpha_0}(t_1) + q \delta_{n(t_1)},$$

or

$$\mathrm{E} \left\| \sum_{i=1}^{n(t_K)} \mathrm{E}(\boldsymbol{\xi}_i \boldsymbol{\xi}_i^\top | \mathcal{F}_{i-1}) - \boldsymbol{V} \right\|_2 \preceq q \delta_{n(t_1)}, \tag{24}$$

since $n^{-\alpha_0}(t_1) \ll \delta_{n(t_1)}$. Combining equation 19 with equation 21, equation 23 and equation 24, an application of Theorem 2.1 in Belloni & Oliveira (2018) yields that

$$|\mathrm{E}\psi(\mathcal{M}_{n(t_K)}) - \mathrm{E}\psi(N(0, \boldsymbol{V}))| \tag{25}$$
$$\preceq c_0(\psi) n^{-\alpha_0}(t_1) + c_2(\psi) q \delta_{n(t_1)} + c_3(\psi) q^{3/2} n^{-1/2}(t_1),$$

for any thrice differential function $\psi(\cdot)$, and

$$c_0(\psi) = \sup_{z,z' \in \mathbb{R}^{pK}} |\psi(z) - \psi(z')| \text{ and } c_i = \sup_{z \in \mathbb{R}^{pK}} \sum_{j_1, \cdots, j_i} |\partial_{j_1} \partial_{j_2} \cdots \partial_{j_i} \psi(z)|, i = 2, 3,$$

where $\partial_j g(z)$ denotes the partial derivative $\partial g(z)/\partial z^{(j)}$ for any function $\mathrm{g}(\cdot)$ and $z^{(j)}$ stands for the $j$-th element of $z$.

Let $\mathbb{X}_{k,0}$ be an $\varepsilon$-net of $\mathbb{X}$ that satisfies the following: for any $x \in \mathbb{X}$, there exists some $x_0 \in \mathbb{X}_0$ such that $\|x - x_0\|_2 \leq \varepsilon$. Set $\varepsilon = \sqrt{d}/n^4(t_1)$. Since $\mathbb{X} = [0,1]^d$, there exists some $\mathbb{X}_0$ with

$$|\mathbb{X}_0| \leq n^{4d}(t_1), \tag{26}$$

where $|\mathbb{X}_0|$ denotes the number of elements in $\mathbb{X}_0$. Under Condition (A3), we have

$$\sup_{x \in \mathbb{X}} \inf_{x_0 \in \mathbb{X}_0} \|\varphi(x) - \varphi(x_0)\|_2 \preceq \frac{\sqrt{q}}{n^4(t_1)}.$$

It follows that

$$\sup_{\|\nu\|_2=1} \left| \sup_{x\in\mathbb{X}} \varphi^\top(x)\nu - \sup_{x\in\mathbb{X}_0} \varphi^\top(x)\nu \right| \preceq \frac{\sqrt{q}}{n^4(t_1)}. \tag{27}$$

Using similar arguments in showing equation 17, we can show the following event occurs with probability at least $1 - O(n^{-1}(t_1))$,

$$\|B^*(t_k)\|_2 \preceq q^{1/2}\log^{1/2} n(t_k), \quad \forall k \geq 1.$$

This together with equation 27 yields

$$\left| \max_{k\in\{1,\ldots,K\}} \sup_{x\in\mathbb{X}} \varphi^\top(x)B^*(t_k) - \max_{k\in\{1,\ldots,K\}} \sup_{x\in\mathbb{X}_0} \varphi^\top(x)B^*(t_k) \right|$$

$$\leq \max_{k\in\{1,\ldots,K\}} \left| \sup_{x\in\mathbb{X}} \varphi^\top(x)B^*(t_k) - \sup_{x\in\mathbb{X}_0} \varphi^\top(x)B^*(t_k) \right| \preceq \frac{q\log n^{1/2}(t_K)}{n^4(t_1)},$$

with probability at least $1 - O(n^{-1}(t_1))$. Under the given conditions, we have $n(t_1) \gg \max(q, \log n(t_K))$. It follows that there exists some constant $\bar{c}^* > 0$ such that

$$\left| \max_{k\in\{1,\ldots,K\}} \sup_{x\in\mathbb{X}} \varphi^\top(x)B^*(t_k) - \max_{k\in\{1,\ldots,K\}} \sup_{x\in\mathbb{X}_0} \varphi^\top(x)B^*(t_k) \right| \leq \bar{c}^* n^{-2}(t_1), \tag{28}$$

with probability at least $1 - O(n^{-1}(t_1))$.

Define

$$z_{k,-}^* = z_k - \bar{c}\{q\delta_{n(t_k)} + \sqrt{qn^{-1}(t_k)\log n(t_k)}\} - \bar{c}^* n^{-2}(t_1),$$
$$z_{k,+}^* = z_k + \bar{c}\{q\delta_{n(t_k)} + \sqrt{qn^{-1}(t_k)\log n(t_k)}\} + \bar{c}^* n^{-2}(t_1).$$

Combining equation 28 with equation 18 yields

$$\Pr\left\{ \max_{k\in\{1,\ldots,K\}} \left( \sup_{x\in\mathbb{X}_0} \varphi^\top(x)B^*(t_k) - z_{k,-}^* \right) \leq 0 \right\} - O(n^{-\alpha_0}(t_1))$$

$$\leq \Pr\left\{ \max_{k\in\{1,\ldots,K\}} \left( \sup_{x\in\mathbb{X}} \varphi^\top(x)B(t_k) - z_k \right) \leq 0 \right\} \tag{29}$$

$$\leq \Pr\left\{ \max_{k\in\{1,\ldots,K\}} \left( \sup_{x\in\mathbb{X}_0} \varphi^\top(x)B^*(t_k) - z_{k,+}^* \right) \leq 0 \right\} + O(n^{-\alpha_0}(t_1)).$$

Notice that $\mathcal{M}_{n(t_K)} = \{B^*(t_1)^\top, B^*(t_2)^\top, \cdots, B^*(t_K)^\top\}^\top$. By equation 26 and Lemma 4, there exist a set of vectors $d_1, d_2, \ldots, d_L \in \mathbb{R}^{qK}$ with $L \leq n^{4d}(t_1)K$, $\max_j \|d_j\|_1 \leq \epsilon_0^{-1} q^{1/2}$ and a function $k(\cdot)$ that maps $\{1,\ldots,L\}$ into $\{1,\ldots,K\}$ such that

$$\max_{k\in\{1,\ldots,K\}} \left\{ \sup_{x\in\mathbb{X}_0} \varphi^\top(x)B^*(t_k) - \nu_k \right\} = \max_{1\leq j\leq L} \{d_j^\top \mathcal{M}_{n(t_K)} - \nu_{k(j)}\}, \tag{30}$$

for any $\{\nu_k\}_{k=1}^K$. For any $\eta > 0$, $m \in \mathbb{R}^{qK}$, consider the function $\phi_{\eta,\{\nu_k\}_k} : \mathbb{R}^{qK} \to \mathbb{R}$, defined as

$$\phi_{\eta,\{\nu_k\}_k}(m) = \frac{1}{\eta}\log\left\{ \sum_{j=1}^L \exp[\eta\{d_j^\top m - \eta\nu_{k(j)}\}] \right\}.$$

It has the following property:

$$\max_{1\leq j\leq L}\{d_j^\top m - \nu_{k(j)}\} \leq \phi_{\eta,\{\nu_k\}_k}(m) \leq \max_{1\leq j\leq L}\{d_j^\top m - \nu_{k(j)}\} + \eta^{-1}\log L$$

$$\leq \max_{1\leq j\leq L}\{d_j^\top m - \nu_{k(j)}\} + \eta^{-1}\{\log K + 4d\log n(t_1)\}$$

$$= \max_{1\leq j\leq L}[d_j^\top m - \{\nu_{k(j)} - \eta^{-1}\log K - \eta^{-1}4d\log n(t_1)\}].$$

It follows that

$$\Pr\left\{\max_{k\in\{1,\dots,K\}}\left(\sup_{x\in\mathbb{X}_0}\varphi^\top(x)B^*(t_k)-z_{k,+}^*\right)\le 0\right\}\le \Pr\left\{\phi_{\eta,\{z_{k,+}^{**}\}_k}(\mathcal{M}_{n(t_K)})\le 0\right\}, \tag{31}$$

$$\Pr\left\{\max_{k\in\{1,\dots,K\}}\left(\sup_{x\in\mathbb{X}_0}\varphi^\top(x)B^*(t_k)-z_{k,-}^*\right)\le 0\right\} \tag{32}$$

$$=\ \Pr\left\{\max_{k\in\{1,\dots,K\}}\left(\sup_{x\in\mathbb{X}_0}\varphi^\top(x)B^*(t_k)-(z_{k,-}^*-3\delta)\right)\le 3\delta\right\}\ge \Pr\left\{\phi_{\eta,\{z_{k,-}^{**}\}_k}(\mathcal{M}_{n(t_K)})\le 3\delta\right\},$$

where

$$z_{k,+}^{**}=z_{k,+}^*+\eta^{-1}\{\log K+4d\log n(t_1)\}\text{ and } z_{k,-}^{**}=z_{k,-}^*-3\delta.$$

The value of $\delta$ will be specified later. In addition, with some calculations, we have

$$\partial_j\phi_{\eta,\{\nu_k\}_k}(m)\ =\ \frac{\sum_{i=1}^L d_i^{(j)}\exp\left(\eta[d_i^\top m-\nu_{k(i)}]\right)}{\sum_{i=1}^L \exp\left(\eta[d_i^\top m-\nu_{k(i)}]\right)},$$

$$\partial_{j_1}\partial_{j_2}\phi_{\eta,\{\nu_k\}_k}(m)\ =\ \eta\frac{\sum_{i=1}^L d_i^{(j_1)}d_i^{(j_2)}\exp\left(\eta[d_i^\top m-\nu_{k(i)}]\right)}{\sum_{i=1}^L \exp\left(\eta[d_i^\top m-\nu_{k(i)}]\right)}$$

$$-\ \eta\frac{\prod_{l=1,2}\left\{\sum_{i=1}^L d_i^{(j_l)}\exp\left(\eta[d_i^\top m-\nu_{k(i)}]\right)\right\}}{\left\{\sum_{i=1}^L \exp\left(\eta[d_i^\top m-\nu_{k(i)}]\right)\right\}^2},$$

$$\partial_{j_1}\partial_{j_2}\partial_{j_3}\phi_{\eta,\{\nu_k\}_k}(m)\ =\ \eta^2\frac{\sum_{i=1}^L d_i^{(j_1)}d_i^{(j_2)}d_i^{(j_3)}\exp\left(\eta[d_i^\top m-\nu_{k(i)}]\right)}{\sum_{i=1}^L \exp\left(\eta[d_i^\top m-\nu_{k(i)}]\right)}$$

$$-\ 3\eta^2\frac{\left\{\sum_{i=1}^L d_i^{(j_1)}d_i^{(j_2)}\exp\left(\eta[d_i^\top m-\nu_{k(i)}]\right)\right\}}{\left\{\sum_{i=1}^L \exp\left(\eta[d_i^\top m-\nu_{k(i)}]\right)\right\}}$$

$$\times\ \frac{\left\{\sum_{i=1}^L d_i^{(j_3)}\exp\left(\eta[d_i^\top m-\nu_{k(i)}]\right)\right\}}{\left\{\sum_{i=1}^L \exp\left(\eta[d_i^\top m-\nu_{k(i)}]\right)\right\}}$$

$$+\ 2\eta^2\frac{\prod_{l=1,2,3}\left(\sum_{i=1}^L d_i^{(j_l)}\exp\left(\eta[d_i^\top m-\nu_{k(i)}]\right)\right)}{\left\{\sum_{i=1}^L \exp\left(\eta[d_i^\top m-\nu_{k(i)}]\right)\right\}^3}.$$

Since $\max_i\|d_i\|_1\le \epsilon_0^{-1}q^{1/2}$, we obtain that

$$\sum_j|\partial_j\phi_{\eta,\{\nu_k\}_k}(m)|\le \epsilon_0^{-1}q^{1/2},\quad \sum_{j_1,j_2}|\partial_{j_1}\partial_{j_2}\phi_{\eta,\{\nu_k\}_k}(m)|\le 2\eta\epsilon_0^{-2}q, \tag{33}$$

$$\sum_{j_1,j_2,j_3}|\partial_{j_1}\partial_{j_2}\partial_{j_3}\phi_{\eta,\{\nu_k\}_k}(m)|\le 6\eta^2\epsilon_0^{-3}q^{3/2}.$$

By Lemma 5.1 of Chernozhukov et al. (2016), for any $\delta>0$, there exists some function $g_\delta(\cdot):\mathbb{R}\to\mathbb{R}$ with $\|g_\delta'\|_\infty\le \delta^{-1}$, $\|g_\delta''\|_\infty\le K_0\delta^{-2}$, $\|g_\delta'''\|_\infty\le K_0\delta^{-3}$ for some constant $K_0>0$ such that

$$\mathbb{I}(z_0\le 0)\le g_\delta(z_0)\le \mathbb{I}(z_0\le 3\delta),\ \forall\delta\in\mathbb{R}.$$

It follows that

$$\mathbb{I}(\phi_{\eta,\{\nu_k\}_k}(m)\le 0)\le g\circ\phi_{\eta,\{\nu_k\}_k}(m)\le \mathbb{I}(\phi_{\eta,\{\nu_k\}_k}(m)\le 3\delta),$$

for any $m\in\mathbb{R}^{qK}$. Combining this together with equation 30, equation 31 and equation 32, we obtain that

$$\Pr\left\{\max_{k\in\{1,\dots,K\}}\left(\sup_{x\in\mathbb{X}_0}\varphi^\top(x)B^*(t_k)-z_{k,+}^*\right)\le 0\right\}\le \mathrm{E}g_\delta\circ\phi_{\eta,\{z_{k,+}^{**}\}_k}(\mathcal{M}_{n(t_K)}), \tag{34}$$

$$\Pr\left\{\max_{k\in\{1,\dots,K\}}\left(\sup_{x\in\mathbb{X}_0}\varphi^\top(x)B^*(t_k)-z_{k,-}^*\right)\le 0\right\}\ge \mathrm{E}g_\delta\circ\phi_{\eta,\{z_{k,-}^{**}\}_k}(\mathcal{M}_{n(t_K)}). \tag{35}$$

Consider the function $g_\delta \circ \phi_{\eta, \{\nu_k\}_k}$. Apparently, we have

$$\sup_{\delta, \eta, \{\nu_k\}_k} c_0(g_\delta \circ \phi_{\eta, \{\nu_k\}_k}) \leq 1. \tag{36}$$

By equation 33, we can show that

$$\sup_{\delta, \eta, \{\nu_k\}_k} c_2(g_\delta \circ \phi_{\eta, \{\nu_k\}_k}) \preceq \delta^{-2}q + \delta^{-1}\eta q,$$
$$\sup_{\delta, \eta, \{\nu_k\}_k} c_3(g_\delta \circ \phi_{\eta, \{\nu_k\}_k}) \preceq \delta^{-3}q^{3/2} + \delta^{-2}\eta q^{3/2} + \delta^{-1}\eta^2 q^{3/2}. \tag{37}$$

Set $\delta = \eta^{-1}\{\log K + 4d \log n(t_1)\}$, we obtain

$$\sup_{\eta, \{\nu_k\}_k} c_i(g_\delta \circ \phi_{\eta, \{\nu_k\}_k}) \preceq q^{i/2}\eta^i \{\log^i K + \log^i n(t_1)\}, \quad i = 2, 3.$$

Combining equation 37 together with equation 25 and equation 36 yields

$$\sup_{\delta, \eta, \{\nu_k\}_k} |Eg_\delta \circ \phi_{\eta, \{\nu_k\}_k}(\mathcal{M}_{n(t_K)}) - Eg_\delta \circ \phi_{\eta, \{\nu_k\}_k}(N(0, \boldsymbol{V}))|$$
$$\preceq n^{-1/2}(t_1)q^3\eta^3\{\log^3 K + \log^3 n(t_1)\} + q^2\eta^2\{\log^2 K + \log^2 n(t_1)\}\delta_{n(t_1)} + n^{-\alpha_0}(t_1).$$

This together with equation 34 and equation 35 yields

$$Pr\left\{\max_{k\in\{1,...,K\}}\left(\sup_{x\in\mathbb{X}_0} \varphi^\top(x)B^*(t_k) - z_{k,+}^*\right) \leq 0\right\} - Eg_\delta \circ \phi_{\eta, \{z_{k,+}^{**}\}_k}(N(0, \boldsymbol{V})) \tag{38}$$
$$\preceq n^{-1/2}(t_1)q^3\eta^3\{\log^3 K + \log^3 n(t_1)\} + q^2\eta^2\{\log^2 K + \log^2 n(t_1)\}\delta_{n(t_1)} + n^{-\alpha_0}(t_1),$$

$$Eg_\delta \circ \phi_{\eta, \{z_{k,-}^{**}\}_k}(N(0, \boldsymbol{V})) - Pr\left\{\max_{k\in\{1,...,K\}}\left(\sup_{x\in\mathbb{X}_0} \varphi^\top(x)B^*(t_k) - z_{k,-}^*\right) \leq 0\right\} \tag{39}$$
$$\preceq n^{-1/2}(t_1)q^3\eta^3\{\log^3 K + \log^3 n(t_1)\} + q^2\eta^2\{\log^2 K + \log^2 n(t_1)\}\delta_{n(t_1)} + n^{-\alpha_0}(t_1).$$

Similar to equation 31-equation 35, we can show

$$Eg_\delta \circ \phi_{\eta, \{z_{k,+}^{**}\}_k}(N(0, \boldsymbol{V})) \leq Pr\left(\phi_{\eta, \{z_{k,+}^{**}\}_k}(N(0, \boldsymbol{V})) \leq 3\delta\right)$$
$$\leq Pr\left(\max_{1\leq j\leq L}\{d_j^\top N(0, \boldsymbol{V}) - z_{k(j),+}^{**}\} \leq 3\delta\right) = Pr\left(\max_{1\leq j\leq L}\{d_j^\top N(0, \boldsymbol{V}) - z_{k(j),+}^{***}\} \leq 0\right),$$

$$Eg_\delta \circ \phi_{\eta, \{z_{k,-}^{**}\}_k}(N(0, \boldsymbol{V})) \geq Pr\left(\phi_{\eta, \{z_{k,-}^{**}\}_k}(N(0, \boldsymbol{V})) \leq 0\right)$$
$$\geq Pr\left(\max_{1\leq j\leq L}\{d_j^\top N(0, \boldsymbol{V}) - z_{k(j),-}^{***}\} \leq 0\right),$$

where

$$z_{k,+}^{***} = z_{k,+}^* + \eta^{-1}\{\log K + 4d \log n(t_1)\} + 3\delta \text{ and } z_{k,-}^{***} = z_{k,-}^* - \eta^{-1}\{\log K + 4d \log n(t_1)\} - 3\delta,$$

for each $k$. Notice that for any $\{\nu_k\}_k$, we have

$$Pr\left\{\max_{k\in\{1,...,K\}}\left(\sup_{x\in\mathbb{X}_0} \varphi^\top(x)G(t_k) - \nu_k\right) \leq 0\right\} = Pr\left(\max_{1\leq j\leq L}\{d_j^\top N(0, \boldsymbol{V}) - \nu_{k(j)}\} \leq 0\right).$$

This together with equation 38 and equation 39 yields

$$Pr\left\{\max_{k\in\{1,...,K\}}\left(\sup_{x\in\mathbb{X}_0} \varphi^\top(x)B^*(t_k) - z_{k,+}^*\right) \leq 0\right\} - Pr\left\{\max_{k\in\{1,...,K\}}\left(\sup_{x\in\mathbb{X}_0} \varphi^\top(x)G(t_k) - z_{k,+}^{***}\right) \leq 0\right\}$$
$$\preceq n^{-1/2}(t_1)q^3\eta^3\{\log^3 K + \log^3 n(t_1)\} + q^2\eta^2\{\log^2 K + \log^2 n(t_1)\}\delta_{n(t_1)} + n^{-\alpha_0}(t_1),$$

$$Pr\left\{\max_{k\in\{1,...,K\}}\left(\sup_{x\in\mathbb{X}_0} \varphi^\top(x)G(t_k) - z_{k,-}^{***}\right) \leq 0\right\} - Pr\left\{\max_{k\in\{1,...,K\}}\left(\sup_{x\in\mathbb{X}_0} \varphi^\top(x)B^*(t_k) - z_{k,-}^*\right) \leq 0\right\}$$
$$\preceq n^{-1/2}(t_1)q^3\eta^3\{\log^3 K + \log^3 n(t_1)\} + q^2\eta^2\{\log^2 K + \log^2 n(t_1)\}\delta_{n(t_1)} + n^{-\alpha_0}(t_1).$$

In view of equation 29, we have shown that

$$
\Pr\left\{ \max_{k\in\{1,...,K\}} \left( \sup_{x\in\mathbb{X}} \varphi^\top(x)B(t_k) - z_k \right) \le 0 \right\} - \Pr\left\{ \max_{k\in\{1,...,K\}} \left( \sup_{x\in\mathbb{X}_0} \varphi^\top(x)G(t_k) - z_{k,+}^{***} \right) \le 0 \right\}
$$
$$
\preceq n^{-1/2}(t_1)q^3\eta^3\{\log^3 K + \log^3 n(t_1)\} + q^2\eta^2\{\log^2 K + \log^2 n(t_1)\}\delta_{n(t_1)} + n^{-\alpha_0}(t_1),
$$
$$
\Pr\left\{ \max_{k\in\{1,...,K\}} \left( \sup_{x\in\mathbb{X}_0} \varphi^\top(x)G(t_k) - z_{k,-}^{***} \right) \le 0 \right\} - \Pr\left\{ \max_{k\in\{1,...,K\}} \left( \sup_{x\in\mathbb{X}} \varphi^\top(x)B(t_k) - z_k \right) \le 0 \right\}
$$
$$
\preceq n^{-1/2}(t_1)q^3\eta^3\{\log^3 K + \log^3 n(t_1)\} + q^2\eta^2\{\log^2 K + \log^2 n(t_1)\}\delta_{n(t_1)} + n^{-\alpha_0}(t_1),
$$

The covariance matrix $\mathrm{Cov}(G(t_k))$ is given by $\sum_{a\in\{0,1\}} \Sigma_a^{-1}\Phi_a\Sigma_a^{-1}$ and is nonsingular by Lemma 4. In addition, we have $\|\varphi(x)\|_2 \ge \bar{c}, \forall x \in \mathbb{X}_0$, by Condition A3. Thus, there exists some constant $c_* > 0$ such that

$$
c_* \le \varphi^\top(x) \left( \sum_{a\in\{0,1\}} \Sigma_a^{-1}\Phi_a\Sigma_a^{-1} \right)^{1/2} \varphi(x), \quad \forall x \in \mathbb{X}_0.
$$

By Theorem 1 of Chernozhukov et al. (2017), we obtain that

$$
\Pr\left\{ \max_{k\in\{1,...,K\}} \left( \sup_{x\in\mathbb{X}_0} \varphi^\top(x)G(t_k) - z_{k,+}^{***} \right) \le 0 \right\} - \Pr\left\{ \max_{k\in\{1,...,K\}} \left( \sup_{x\in\mathbb{X}_0} \varphi^\top(x)G(t_k) - z_{k,-}^{***} \right) \le 0 \right\}
$$
$$
\preceq \eta^{-1}\{\log K + \log n(t_1)\}^{3/2} + q\delta_{n(t_1)}\{\log K + \log n(t_1)\}^{1/2} + \sqrt{qn^{-1}(t_1)\log n(t_1)}\{\log K + \log n(t_1)\}^{1/2}.
$$

Thus, we obtain

$$
\left| \Pr\left\{ \max_{k\in\{1,...,K\}} \left( \sup_{x\in\mathbb{X}} \varphi^\top(x)B(t_k) - z_k \right) \le 0 \right\} - \Pr\left\{ \max_{k\in\{1,...,K\}} \left( \sup_{x\in\mathbb{X}} \varphi^\top(x)G(t_k) - z_k \right) \le 0 \right\} \right|
$$
$$
\preceq n^{-1/2}(t_1)q^3\eta^3\{\log^3 K + \log^3 n(t_1)\} + q^2\eta^2\{\log^2 K + \log^2 n(t_1)\}\delta_{n(t_1)} + n^{-\alpha_0}(t_1)
$$
$$
+ \eta^{-1}\{\log K + \log n(t_1)\}^{3/2} + q\delta_{n(t_1)}\{\log K + \log n(t_1)\}^{1/2} + \sqrt{qn^{-1}(t_1)\log n(t_1)}\{\log K + \log n(t_1)\}^{1/2}.
$$

Setting $\eta = \min(q^{-3/4}n^{1/8}(t_1)\log^{-3/8}\{Kn(t_1)\}, q^{-1}n^{-\alpha_0/3}(t_1)\log^{-\alpha_0/3-1/6}\{Kn(t_1)\})$ yields the desired results. The proof is hence completed.

### E.3 PROOF OF LEMMA 3

The assertion trivially holds for $j = 1$. We prove it holds for any $j \ge 2$, by induction. By (A2), we have $(X_j, Y_j^*(0), Y_j^*(1)) \perp\!\!\!\perp A_1 | X_1$. Since $(X_j, Y_j^*(0), Y_j^*(1)) \perp\!\!\!\perp (X_1, Y_1^*(0), Y_1^*(1))$, this further implies $(X_j, Y_j^*(0), Y_j^*(1)) \perp\!\!\!\perp A_1$ and hence $(X_j, Y_j^*(0), Y_j^*(1)) \perp\!\!\!\perp (X_1, A_1, Y_1^*(0), Y_1^*(1))$. By (A1), $Y_1$ is completely determined by $A_1$, $Y_1^*(0)$ and $Y_1^*(1)$. Therefore, we obtain $(X_j, Y_j^*(0), Y_j^*(1)) \perp\!\!\!\perp \mathcal{F}_1$.

Suppose we have shown that $(X_j, Y_j^*(0), Y_j^*(1)) \perp\!\!\!\perp \mathcal{F}_k$ for some $k < j - 1$. To prove $(X_j, Y_j^*(0), Y_j^*(1)) \perp\!\!\!\perp \mathcal{F}_{k+1}$, it suffices to show $(X_j, Y_j^*(0), Y_j^*(1)) \perp\!\!\!\perp (X_{k+1}, A_{k+1}, Y_{k+1})$. By (A1), $Y_{k+1}$ is determined by $A_{k+1}$, $Y_{k+1}^*(0)$ and $Y_{k+1}^*(1)$. Since $(X_j, Y_j^*(0), Y_j^*(1)) \perp\!\!\!\perp (X_{k+1}, Y_{k+1}^*(0), Y_{k+1}^*(1))$, it suffices to show $(X_j, Y_j^*(0), Y_j^*(1)) \perp\!\!\!\perp A_{k+1}$. This is implied by $(X_j, Y_j^*(0), Y_j^*(1)) \perp\!\!\!\perp A_{k+1} | X_{k+1}, \mathcal{F}_k$ and that $(X_j, Y_j^*(0), Y_j^*(1)) \perp\!\!\!\perp X_{k+1}, \mathcal{F}_k$. The proof is hence completed.

### E.4 PROOF OF LEMMA 4

The assertions

$$
\epsilon_0 \le \lambda_{\min}[\mathrm{E}\varphi(X)\varphi^\top(X)] \le \lambda_{\max}[\mathrm{E}\varphi(X)\varphi^\top(X)] \le \epsilon_0^{-1}, \tag{40}
$$

and

$$
\sup_x \|\varphi(x)\|_1 \le \epsilon_0^{-1}\sqrt{q}, \tag{41}
$$

for some $0 < \epsilon_0 < 1$ are directly implied by the conditions that $\lambda_{\min}[\mathrm{E}\varphi(X)\varphi^\top(X)] \asymp 1$, $\lambda_{\max}[\mathrm{E}\varphi(X)\varphi^\top(X)] \asymp 1$, $\sup_x \|\varphi(x)\|_1 \leq \epsilon_0^{-1}\sqrt{q}$. Since $\|\varphi(x)\|_2 \leq \|\varphi(x)\|_1$, we obtain $\sup_x \|\varphi(x)\|_2 \leq \sup_x \|\varphi(x)\|_1 \leq \epsilon_0^{-1}\sqrt{q}$.

Under the condition $\inf_{a,x} \pi^*(a,x) > 0$, we can similarly show that $\lambda_{\min}[\Sigma_a] \geq \epsilon_0$ for some $\epsilon_0 > 0$.

Since $|Y^*(0)|$ and $|Y^*(1)|$ are bounded, there exists some constant $0 < \epsilon_0 < 1$ that satisfies $\max_{a \in \{0,1\}} |Y^*(a)| \leq \epsilon_0^{-1}$. Notice that $\varphi^\top(x)\beta_a = \mathrm{E}\{Y^*(a)|X = x\}$. Boundedness of $|Y^*(a)|$ implies that the conditional mean $\mathrm{E}\{Y^*(a)|X\}$ is a bounded random variable as well. As a result, we obtain $\sup_{x \in \mathbb{X}} \max_{a \in \{0,1\}} |\varphi^\top(x)\beta_a| \leq \epsilon_0^{-1}$.

Notice that $\beta_a = \Sigma_a^{-1}\mathrm{E}\varphi^\top(X)Y^*(a)$. Since $\lambda_{\min}[\Sigma_a]$ is bounded away from 0, it suffices to show $\|\mathrm{E}\varphi^\top(X)Y^*(a)\|_2 = O(1)$, or equivalently,

$$\sup_{\nu \in \mathbb{R}^p, \|\nu\|_2 = 1} |\mathrm{E}\nu^\top \varphi(X)Y^*(a)| = O(1).$$

By Cauchy-Schwarz inequality, it suffices to show

$$\sup_{\nu \in \mathbb{R}^p, \|\nu\|_2 = 1} \mathrm{E}|Y^*(a)|^2 \mathrm{E}|\nu^\top \varphi(X)|^2 = O(1).$$

Since $|Y^*(a)| = O(1)$ almost surely, we have by the condition $\lambda_{\max}[\mathrm{E}\varphi(X)\varphi^\top(X)] = O(1)$ that

$$\sup_{\nu \in \mathbb{R}^p, \|\nu\|_2 = 1} \mathrm{E}|\nu^\top \varphi(X)|^2 = \sup_{\nu \in \mathbb{R}^p, \|\nu\|_2 = 1} \nu^\top \mathrm{E}\varphi(X)\varphi^\top(X)\nu \leq \lambda_{\max}[\mathrm{E}\varphi(X)\varphi^\top(X)] = O(1).$$

The proof is hence completed.

### E.5  PROOF OF LEMMA 5

#### E.5.1  PROOF OF EQUATION 14

Notice that

$$\|j(\widehat{\Sigma}_{1,j} - \Sigma_1)\|_2 = \left\|\sum_{i=1}^{j}\{A_i\varphi(X_i)\varphi^\top(X_i) - \mathrm{E}^{\mathcal{F}_{i-1}}\pi_{i-1}(1,X)\varphi(X)\varphi^\top(X)\}\right\|_2 \tag{42}$$

$$+ j\left\|\mathrm{E}^{\mathcal{F}_{i-1}}\varphi(X)\varphi^\top(X)\left(\frac{1}{j}\sum_{i=1}^{j}\pi_{i-1}(1,X) - \pi^*(1,X)\right)\right\|_2.$$

By Lemma 4, we have

$$\left\|\mathrm{E}^{\mathcal{F}_{i-1}}\varphi(X)\varphi^\top(X)\left(\frac{1}{j}\sum_{i=1}^{j}\pi_{i-1}(1,X) - \pi^*(1,X)\right)\right\|_2 \leq \varepsilon_0^{-2}q\mathrm{E}^{\mathcal{F}_{i-1}}\left|\frac{1}{j}\sum_{i=1}^{j}\pi_{i-1}(1,X) - \pi^*(1,X)\right|$$

$$\leq \varepsilon_0^{-2}q^2 j^{-\alpha_0}\log^{\alpha_0} j, \quad \forall j \geq j_n,$$

with probability at least $1 - O(j_n^{-\alpha_0})$.

Consider the first term on the RHS of equation 42. For any $i \geq 1$, define $M_i = \varphi(X_i)\varphi^\top(X_i)\{A_i - \pi_{i-1}(1,X_i)\}$. Notice that $\{M_i\}_{i \geq 1}$ forms a martingale difference sequence with respect to the filtration $\{\sigma(\mathcal{F}_{i-1}) : i \geq 2\}$, since

$$\mathrm{E}[\varphi(X_i)\varphi^\top(X_i)\{A_i - \pi_{i-1}(X_i)\}|\mathcal{F}_{i-1}] \tag{43}$$

$$= \mathrm{E}^{\mathcal{F}_{i-1}}[\mathrm{E}(\varphi(X_i)\varphi(X_i)^\top\{A_i - \pi_{i-1}(X_i)\}|\mathcal{F}_{i-1}, X_i)] = 0,$$

where $\mathrm{E}^{\mathcal{F}_i, X_i}$ denotes the conditional expectation given $X_i$ and $\mathcal{F}_i$. Here, the first equality is due to that $X_i \perp\!\!\!\perp \mathcal{F}_{i-1}$, implied by Lemma 3. Under the given conditions on the basis function $\varphi(\cdot)$, using similar arguments in proving Equation (C.15) of Shi et al. (2020b), we can show that the following event occurs with probability at least $1 - O(j^{-2})$,

$$\left\|\sum_{i=1}^{j} M_i\right\|_2 \preceq \sqrt{qj\log(j)}.$$

Notice that $\sum_{k \geq j} k^{-2} \leq j^{-2} + \sum_{k > j} \{k(k-1)\}^{-1} = j^{-2} + j^{-1}$. Thus, the following occurs with probability at least $1 - O(j_n^{-1})$,

$$\left\| \sum_{i=1}^{j} \{A_i \varphi(X_i) \varphi^\top(X_i) - \mathrm{E}^{\mathcal{F}_{i-1}} \pi_{i-1}(1, X) \varphi(X) \varphi^\top(X)\} \right\|_2 \preceq \sqrt{qj \log j}, \quad \forall j \geq j_n. \tag{44}$$

It follows that

$$\|(\widehat{\Sigma}_{1,k} - \Sigma_1)\|_2 \preceq q\delta_k + \sqrt{qk^{-1} \log k}, \quad \forall k \geq j_n,$$

with probability at least $1 - O(j^{-\alpha_0})$. Similarly, we can show

$$\|(\widehat{\Sigma}_{0,k} - \Sigma_0)\|_2 \preceq q\delta_k + \sqrt{qk^{-1} \log k}, \quad \forall k \geq j_n,$$

with probability at least $1 - O(j_n^{-\alpha_0})$. The proof is hence completed.

### E.5.2 PROOF OF EQUATION 15

When $j_n$ satisfies $j_n^{\alpha_0} / \log^{\alpha_0}(j_n) \gg q^2$, it follows from equation 14 and equation 40 that

$$\lambda_{\min}[\widehat{\Sigma}_{a,k}] \geq \lambda_{\min}[\Sigma_a] - \|\widehat{\Sigma}_{a,k} - \Sigma_a\|_2 \geq 2^{-1}\varepsilon_0, \quad \forall k \geq j_n, \tag{45}$$

with probability at least $1 - O(j_n^{-\alpha_0})$. Combining equation 40 with equation 45 and equation 14, we obtain

$$\|\widehat{\Sigma}_{a,k}^{-1} - \Sigma_a^{-1}\|_2 = \|\widehat{\Sigma}_{a,k}^{-1}(\widehat{\Sigma}_{a,k} - \Sigma_a)\Sigma_a^{-1}\|_2 \leq \lambda_{\min}[\Sigma_a]\lambda_{\min}[\widehat{\Sigma}_{a,k}]\|\widehat{\Sigma}_{a,k} - \Sigma_a\|_2$$
$$\preceq q\delta_k + \sqrt{qk^{-1} \log k}, \quad \forall k \geq j_n,$$

with probability at least $1 - O(j_n^{-\alpha_0})$. The proof is hence completed.

### E.6 PROOF OF LEMMA 6

For any $l \in \{1, \ldots, q\}$ and $i \geq 1$, define $M_i(l) = \varphi^{(l)}(X_i) A_i \{Y_i - \varphi^\top(X_i)\beta_1\}$. Here, $\varphi^{(l)}(X_i)$ corresponds to the $l$-th element of $\varphi(X_i)$. Similar to equation 43, we can show $\{M_i(l)\}_{i \geq 1}$ forms a martingale difference sequence with respect to the filtration $\{\sigma(\mathcal{F}_{i-1}) : i \geq 1\}$. By equation 43, we have for any $l$,

$$\mathrm{E}\{\varphi^{(l)}(X_i)\}^2 \leq \lambda_{\max}[\varphi(X_i)\varphi^\top(X_i)] \leq \epsilon_0^{-1}. \tag{46}$$

Notice that

$$\mathrm{E}\{M_i^2(l)|\mathcal{F}_{i-1}\} = \mathrm{E}[\{\varphi^{(l)}(X_i)\}^2 A_i \{Y_i^*(1) - \varphi^\top(X_i)\beta_1\}^2 | \mathcal{F}_{i-1}]$$
$$\leq \mathrm{E}[\{\varphi^{(l)}(X_i)\}^2 \{Y_i^*(1) - \varphi^\top(X_i)\beta_1\}^2 | \mathcal{F}_{i-1}] = \mathrm{E}\sigma^2(1, X_i)\{\varphi^{(l)}(X_i)\}^2$$
$$\leq 4\epsilon_0^{-2} \mathrm{E}\{\varphi^{(l)}(X_i)\}^2 \leq 4\epsilon_0^{-3},$$

where the first equality is due to (A1), the first inequality is due to that $A$ is bounded between $0$ and $1$, the second equality follows from Lemma 3, the second inequality follows from Lemma 4, and the last inequality is due to equation 46. It follows that

$$\sum_{i=1}^{k} \mathrm{E}\{M_i^2(l)|\mathcal{F}_{i-1}\} \leq 4k\epsilon_0^{-3}. \tag{47}$$

Similarly, by (A1) and Lemma 4, we have

$$\sum_{i=1}^{k} M_i^2(l) \leq 4\epsilon_0^{-2} \sum_{i=1}^{k} \{\psi^{(l)}(X_i)\}^2. \tag{48}$$

Similar to equation 44, we can show with probability at least $1 - O(j^{-1})$ that

$$\sum_{i=1}^{k} [M_i^2(l) - \mathrm{E}\{M_i^2(l)|\mathcal{F}_{i-1}\}] \preceq \sqrt{qk \log k}, \quad \forall k \geq j.$$

Thus, for any sequence $j_n$ that satisfies $j_n/\log(j_n) \gg q$, we have by equation 47 that

$$\sum_{i=1}^{k} M_i^2(l) + \sum_{i=1}^{k} \mathrm{E}\{M_i^2(l)|\mathcal{F}_{i-1}\} \le \bar{c}k, \quad \forall k \ge j_n,$$

for some constant $\bar{c} > 0$, with probability at least $1 - O(j_n^{-1})$. It follows that

$$\begin{aligned}
&\Pr\left(\bigcap_{k \ge j_n} \{|\sum_{i=1}^{k} M_i(l)| \le 2\sqrt{\bar{c}k\log k}\}\right) \\
\ge\ &\Pr\left(\left\{\bigcap_{k \ge j_n}\{|\sum_{i=1}^{k} M_i(l)| \le 2\sqrt{\bar{c}k\log k}\}\right\} \cap \left\{\bigcap_{k \ge j_n}\{\sum_{i=1}^{k}[M_i^2(l) + \{M_i^2(l)|\mathcal{F}_{i-1}\}] \le \bar{c}k\}\right\}\right) \\
-\ &O(j_n^{-1}) \ge \Pr\left(\left\{\bigcap_{k \ge j_n}\{\sum_{i=1}^{k}[M_i^2(l) + \{M_i^2(l)|\mathcal{F}_{i-1}\}] \le \bar{c}k\}\right\}\right) - O(j_n^{-1}) \\
-\ &\Pr\left(\left\{\bigcup_{k \ge j_n}\{|\sum_{i=1}^{k} M_i(l)| > 2\sqrt{\bar{c}k\log k}\}\right\} \cap \left\{\bigcap_{k \ge j_n}\{\sum_{i=1}^{k}[M_i^2(l) + \{M_i^2(l)|\mathcal{F}_{i-1}\}] \le \bar{c}k\}\right\}\right) \\
\ge\ &1 - \Pr\left(\left\{\bigcup_{k \ge j_n}\{|\sum_{i=1}^{k} M_i(l)| > 2\sqrt{\bar{c}k\log k}\}\right\} \cap \left\{\bigcap_{k \ge j_n}\{\sum_{i=1}^{k}[M_i^2(l) + \{M_i^2(l)|\mathcal{F}_{i-1}\}] \le \bar{c}k\}\right\}\right) \\
-\ &O(j_n^{-1}).
\end{aligned}$$

By Bonferroni's inequality and Theorem 2.1 of Bercu & Touati (2008), we have

$$\begin{aligned}
&\Pr\left(\bigcap_{k \ge j_n}\{|\sum_{i=1}^{k} M_i(l)| \le 2\sqrt{\bar{c}k\log k}\}\right) \ge 1 - O(j_n^{-1}) \\
-\ &\sum_{k \ge j_n}\Pr\left(\{|\sum_{i=1}^{k} M_i(l)| > 2\sqrt{\bar{c}k\log k}\} \cap \left\{\bigcap_{k' \ge j_n}\{\sum_{i=1}^{k'}[M_i^2(l) + \{M_i^2(l)|\mathcal{F}_{i-1}\}] \le \bar{c}k'\}\right\}\right) \\
\ge\ &1 - O(j_n^{-1}) - \sum_{k \ge j_n}\Pr\left(\{|\sum_{i=1}^{k} M_i(l)| > 2\sqrt{\bar{c}k\log k}\} \cap \left\{\sum_{i=1}^{k}[M_i^2(l) + \{M_i^2(l)|\mathcal{F}_{i-1}\}] \le \bar{c}k\right\}\right) \\
\ge\ &1 - O(j_n^{-1}) - 2\sum_{k \ge j_n}\exp\left(-\frac{4\bar{c}k\log k}{2\bar{c}k}\right) = 1 - O(j_n^{-1}) - \sum_{k \ge j_n}2k^{-2}. \qquad (49)
\end{aligned}$$

The last term on the RHS of equation 49 is $1 - O(j_n^{-1})$. To summarize, we have shown that the following event occurs with probability at least $1 - O(j_n^{-1})$,

$$\bigcap_{k \ge j_n}\left\{|\sum_{i=1}^{k} M_i(l)| \le 2\sqrt{\bar{c}k\log k}\right\}.$$

By Bonferroni's inequality, we have

$$\bigcap_{k \ge j_n}\left\{\left\|\sum_{i=1}^{k}\varphi(X_i)A_i\{Y_i - \varphi^\top(X_i)\beta_1\}\right\|_2 \le 2\sqrt{\bar{c}qk\log k}\right\},$$

with probability at least $1 - O(j_n^{-1/2})$. Similarly, we can show

$$\bigcap_{k \ge j_n}\left\{\left\|\sum_{i=1}^{k}\varphi(X_i)(1 - A_i)\{Y_i - \varphi^\top(X_i)\beta_0\}\right\|_2 \le c\sqrt{qk\log k}\right\},$$

for some constant $c > 0$, with probability at least $1 - O(j_n^{-1})$. The proof is hence completed.

### E.7 PROOF OF THEOREM 3

We state the following lemmas before presenting the proof.

**Lemma 7** *Assume the conditions in Theorem 3 hold. Then for any sequence $\{j_n\}_n$ that satisfies $j_n^{\alpha_0}/\log^{\alpha_0} j_n \gg q^2$, we have with probability at least $1 - O(j_n^{-\alpha_0})$ that*

$$\|\widehat{\beta}_{a,k} - \beta_a\|_2 \preceq q^{1/2} k^{-1/2} \sqrt{\log k}, \quad \forall a \in \{0, 1\}, \forall k \geq j_n.$$

**Lemma 8** *Assume the conditions in Theorem 3 hold. Then for any sequence $\{j_n\}_n$ that satisfies $j_n^{\alpha_0}/\log^{\alpha_0} j_n \gg q^2$, we have with probability at least $1 - O(j_n^{-\alpha_0})$ that*

$$\left\| \frac{1}{k} \sum_{i=1}^{k} \mathbb{I}(A_i = a)\varphi(X_i)\varphi^\top(X_i)\{Y_i - \varphi^\top(X_i)\beta_a\}^2 - \Phi_a \right\|_2 \preceq q\delta_k + q^{1/2}k^{-1/2}\sqrt{\log k},$$

$$\forall a \in \{0, 1\}, k \geq j_n.$$

Similar to the proof of Theorem 1, we will show the assertion in Theorem 3 holds for any $n(\cdot)$ that correspond to the realizations of $N(\cdot)$ that satisfy $n(t_1) < n(t_2) < \cdots < n(t_K)$. For any $1 \leq k_1 \leq k_2 \leq K$, define

$$\widehat{V}(k_1, k_2) = \sqrt{n(t_{k_1})n(t_{k_2})}\text{Cov}\left(\widehat{\beta}_1^{\text{MB}*}(t_{k_1}) - \widehat{\beta}_0^{\text{MB}*}(t_{k_1}), \widehat{\beta}_1^{\text{MB}*}(t_{k_2}) - \widehat{\beta}_0^{\text{MB}*}(t_{k_2})|\{(X_i, A_i, Y_i)\}_{i=1}^{+\infty}\right)$$

$$= \frac{1}{\sqrt{n(t_{k_1})n(t_{k_2})}} \sum_{a=0}^{1} \sum_{j=1}^{k_1} \sum_{i=n(t_{j-1})+1}^{n(t_j)} \widehat{\Sigma}_a^{-1}(t_j)\mathbb{I}(A_i = a)\varphi(X_i)\varphi^\top(X_i)\{Y_i - \varphi^\top(X_i)\widehat{\beta}_a(t_j)\}^2\widehat{\Sigma}_a^{-1}(t_j),$$

and

$$\widehat{V} = \begin{pmatrix} \widehat{V}(1,1) & \widehat{V}(1,2) & \ldots & \widehat{V}(1,K) \\ \widehat{V}(2,1) & \widehat{V}(2,2) & \ldots & \widehat{V}(2,K) \\ \vdots & \vdots & & \vdots \\ \widehat{V}(K,1) & \widehat{V}(K,2) & \ldots & \widehat{V}(K,K) \end{pmatrix}.$$

We aim to bound the entrywise $\ell_\infty$ norm of $\widehat{V} - V$ where $V$ is defined in equation 22. It suffices to bound $\max_{1 \leq k_1 \leq k_2 \leq K} \sup_{b_1, b_2 \in \mathbb{R}^{p+1}, \|b_1\|_2 = \|b_2\|_2 = 1} |b_1^T \{\widehat{V}(k_1, k_2) - V(k_1, k_2)\}b_2| = \max_{1 \leq k_1 \leq k_2 \leq K} \|\widehat{V}(k_1, k_2) - V(k_1, k_2)\|_2$. For any $k_1, k_2$, we decompose $\widehat{V}(k_1, k_2) - V(k_1, k_2)$ as

$$\widehat{V}(k_1, k_2) - V(k_1, k_2) = \widehat{V}(k_1, k_2) - \widehat{V}^*(k_1, k_2) + \widehat{V}^*(k_1, k_2) - \widehat{V}^{**}(k_1, k_2) + \widehat{V}^{**}(k_1, k_2) - V(k_1, k_2),$$

where

$$\widehat{V}^*(k_1, k_2) = \frac{1}{\sqrt{n(t_{k_1})n(t_{k_2})}} \sum_{a=0}^{1} \sum_{j=1}^{k_1} \sum_{i=n(t_{j-1})+1}^{n(t_j)} \Sigma_a^{-1}\mathbb{I}(A_i = a)\varphi(X_i)\varphi^\top(X_i)\{Y_i - \varphi^\top(X_i)\widehat{\beta}_a(t_j)\}^2\Sigma_a^{-1},$$

$$\widehat{V}^{**}(k_1, k_2) = \frac{1}{\sqrt{n(t_{k_1})n(t_{k_2})}} \sum_{a=0}^{1} \sum_{j=1}^{n(t_{k_1})} \Sigma_a^{-1}\mathbb{I}(A_i = a)\varphi(X_i)\varphi^\top(X_i)\{Y_i - \varphi^\top(X_i)\beta_a\}^2\Sigma_a^{-1}.$$

By Lemma 4 and Lemma 8, we obtain that

$$\max_{1 \leq k_1 \leq k_2 \leq K} \|\widehat{V}^{**}(k_1, k_2) - V(k_1, k_2)\|_2$$

$$\leq \max_{1 \leq k_1 \leq K} \sum_{a=0}^{1} \left\| \frac{1}{n(t_{k_1})} \sum_{j=1}^{n(t_{k_1})} \Sigma_a^{-1}\mathbb{I}(A_i = a)\varphi(X_i)\varphi^\top(X_i)\{Y_i - \varphi^\top(X_i)\beta_a\}^2\Sigma_a^{-1} - \Sigma_a^{-1}\Phi_a\Sigma_a^{-1} \right\|_2$$

$$\leq \max_{1 \leq k_1 \leq K} \frac{1}{\epsilon_0^2} \sum_{a=0}^{1} \left\| \frac{1}{n(t_{k_1})} \sum_{j=1}^{n(t_{k_1})} \mathbb{I}(A_i = a)\varphi(X_i)\varphi^\top(X_i)\{Y_i - \varphi^\top(X_i)\beta_a\}^2 - \Phi_a \right\|_2$$

$$\preceq q\delta_{n(t_1)} + q^{1/2}n^{-1/2}(t_1)\sqrt{\log n(t_1)}, \tag{50}$$

with probability at least $1 - O(n^{-\alpha_0}(t_1))$.

Notice that

$$\sqrt{n(t_{k_1})n(t_{k_2})}\widehat{V}^*(k_1, k_2) = \sum_{a=0}^{1}\sum_{j=1}^{k_1}\sum_{i=n(t_{j-1})+1}^{n(t_j)} \Sigma_a^{-1}\mathbb{I}(A_i = a)\varphi(X_i)\varphi^\top(X_i)\{Y_i - \varphi^\top(X_i)\widehat{\beta}_a(t_j)\}^2\Sigma_a^{-1}$$

$$= \sum_{a=0}^{1}\sum_{j=1}^{k_1}\sum_{i=n(t_{j-1})+1}^{n(t_j)} \Sigma_a^{-1}\mathbb{I}(A_i = a)\varphi(X_i)\varphi^\top(X_i)\{Y_i - \varphi^\top(X_i)\beta_a + \varphi^\top(X_i)\beta_a - \varphi^\top(X_i)\widehat{\beta}_a(t_j)\}^2\Sigma_a^{-1}$$

$$= \sum_{a=0}^{1}\sum_{j=1}^{k_1}\sum_{i=n(t_{j-1})+1}^{n(t_j)} \Sigma_a^{-1}\mathbb{I}(A_i = a)\varphi(X_i)\varphi^\top(X_i)\{\varphi^\top(X_i)\beta_a - \varphi^\top(X_i)\widehat{\beta}_a(t_j)\}^2\Sigma_a^{-1}$$

$$+ 2\sum_{a=0}^{1}\sum_{j=1}^{k_1}\sum_{i=n(t_{j-1})+1}^{n(t_j)} \Sigma_a^{-1}\mathbb{I}(A_i = a)\varphi(X_i)\varphi^\top(X_i)\{Y_i - \varphi^\top(X_i)\beta_a\}\varphi^\top(X_i)\{\beta_a - \widehat{\beta}_a(t_j)\}\Sigma_a^{-1}$$

$$+ \sqrt{n(t_{k_1})n(t_{k_2})}\widehat{V}^{**}(k_1, k_2).$$

It follows that

$$\max_{1\le k_1 \le k_2 \le K} \left\|\widehat{V}^*(k_1, k_2) - \widehat{V}^{**}(k_1, k_2)\right\|_2$$

$$\le \max_{1\le k_1 \le K} \frac{1}{n(t_{k_1})} \left\| \sum_{a=0}^{1}\sum_{j=1}^{k_1}\sum_{i=n(t_{j-1})+1}^{n(t_j)} \Sigma_a^{-1}\mathbb{I}(A_i = a)\varphi(X_i)\varphi^\top(X_i)\{\varphi^\top(X_i)\beta_a - \varphi^\top(X_i)\widehat{\beta}_a(t_j)\}^2\Sigma_a^{-1}\right\|_2$$

$$+ \max_{1\le k_1 \le K} \frac{2}{n(t_{k_1})} \left\| \sum_{a=0}^{1}\sum_{j=1}^{k_1}\sum_{i=n(t_{j-1})+1}^{n(t_j)} \Sigma_a^{-1}\mathbb{I}(A_i = a)\varphi(X_i)\varphi^\top(X_i)\{Y_i - \varphi^\top(X_i)\beta_a\}\varphi^\top(X_i)(\beta_a - \widehat{\beta}_a(t_j))\Sigma_a^{-1}\right\|_2.$$

By Lemma 4, we obtain that

$$\max_{1\le k_1 \le k_2 \le K} \left\|\widehat{V}^*(k_1, k_2) - \widehat{V}^{**}(k_1, k_2)\right\|_2 \tag{51}$$

$$\preceq \max_{\substack{1\le k_1 \le K \\ a \in \{0,1\}}} \frac{1}{n(t_{k_1})} \left\| \underbrace{\sum_{j=1}^{k_1}\sum_{i=n(t_{j-1})+1}^{n(t_j)} \mathbb{I}(A_i = a)\varphi(X_i)\varphi^\top(X_i)\{\varphi^\top(X_i)\beta_a - \varphi^\top(X_i)\widehat{\beta}_a(t_j)\}^2}_{\Psi_{1,a,k_1}}\right\|_2$$

$$+ \max_{\substack{1\le k_1 \le K \\ a \in \{0,1\}}} \frac{2}{n(t_{k_1})} \left\| \underbrace{\sum_{j=1}^{k_1}\sum_{i=n(t_{j-1})+1}^{n(t_j)} \mathbb{I}(A_i = a)\varphi(X_i)\varphi^\top(X_i)\{Y_i - \varphi^\top(X_i)\beta_a\}\varphi^\top(X_i)\{\beta_a - \widehat{\beta}_a(t_j)\}}_{\Psi_{2,a,k_1}}\right\|_2.$$

By Lemmas 4 and 7, we have with probability at least $1 - O(n^{-1}(t_1))$ that

$$\frac{1}{n(t_{k_1})}\|\Psi_{1,a,k_1}\|_2 \preceq q^2 n^{-1}(t_1)\log\{n(t_1)\} \left\| \frac{1}{n(t_{k_1})}\sum_{i=1}^{n(t_{k_1})}\mathbb{I}(A_i = a)\varphi(X_i)\varphi^\top(X_i)\right\|_2, \tag{52}$$

$$\forall 1 \le k_1 \le K, a \in \{0, 1\}.$$

Similar to Lemma 5, we can show there exists some constant $c_* > 0$ that

$$\frac{1}{n(t_{k_1})} \left\| \sum_{i=1}^{n(t_{k_1})}[\mathbb{I}(A_i = a)\varphi(X_i)\varphi^\top(X_i) - \mathrm{E}^{\mathcal{F}_{i-1}}\{\mathbb{I}(A_i = a)\varphi(X_i)\varphi^\top(X_i)\}]\right\|_2 \tag{53}$$

$$\le c_*\{q\delta_{n(t_{k_1})} + q^{1/2}n^{-1/2}(t_{k_1})\sqrt{\log n(t_{k_1})}\}, \quad \forall 1 \le k_1 \le K, a \in \{0, 1\},$$

with probability at least $1 - O(n^{-1}(t_1))$. By Lemma 4, we can show with probability at least $1 - O(n^{-1}(t_1))$ that

$$\max_{1 \le k_1 \le K} \frac{1}{n(t_{k_1})} \left\| \sum_{i=1}^{n(t_{k_1})} \mathrm{E}^{\mathcal{F}_{i-1}} \{ \mathbb{I}(A_i = a) \varphi(X_i) \varphi^\top(X_i) \} \right\|_2 = O(1).$$

This together with equation 52 and equation 53 yields

$$n^{-1}(t_{k_1}) \| \Psi_{1,a,k_1} \|_2 \preceq q^2 n^{-1}(t_1) \log\{n(t_1)\}, \quad \forall 1 \le k_1 \le K, a \in \{0, 1\}, \tag{54}$$

with probability at least $1 - O(n^{-1}(t_1))$.

Moreover, using similar arguments in proving Equation (C.15) of Shi et al. (2020b), we can show that for any $1 \le k_1 \le K$, the following event occurs with probability at least $1 - O(n^{-2}(t_{k_1}))$,

$$\frac{1}{n(t_{k_1})} \left\| \sum_{i=1}^{n(t_{k_1})} \mathbb{I}(A_i = a) \varphi(X_i) \varphi^\top(X_i) \{Y_i - \varphi^\top(X_i)\beta_a\} \varphi^{(l)}(X_i) \right\|_2 \preceq q^{1/2} n^{-1/2}(t_{k_1}) \sqrt{\log n(t_{k_1})},$$

$$\forall 1 \le l \le q.$$

Since $\sum_{k_1=1}^{K} n^{-2}(t_{k_1}) \le n^{-1}(t_1)$, we obtain with probability at least $1 - O(n^{-1}(t_1))$ that

$$\frac{1}{n(t_{k_1})} \left\| \sum_{i=1}^{n(t_{k_1})} \mathbb{I}(A_i = a) \varphi(X_i) \varphi^\top(X_i) \{Y_i - \varphi^\top(X_i)\beta_a\} \varphi^{(l)}(X_i) \right\|_2 \preceq q^{1/2} n^{-1/2}(t_{k_1}) \sqrt{\log n(t_{k_1})},$$

$$\forall 1 \le l \le q, 1 \le k_1 \le K.$$

In addition, it follows from Lemma 7 that

$$n^{-1}(t_{k_1}) \| \Psi_{2,a,k_1} \|_2 \preceq q^{3/2} n^{-1}(t_1) \log\{n(t_1)\}, \quad \forall 1 \le k_1 \le K, a \in \{0, 1\}.$$

This together with equation 54 yields that

$$\max_{1 \le k_1 \le k_2 \le K} \left\| \widehat{V}^*(k_1, k_2) - \widehat{V}^{**}(k_1, k_2) \right\|_2 \preceq q^2 n^{-1}(t_1) \log n(t_1),$$

with probability at least $1 - O(n^{-1}(t_1))$. Under the given conditions, we have

$$\max_{1 \le k_1 \le k_2 \le K} \left\| \widehat{V}^*(k_1, k_2) - \widehat{V}^{**}(k_1, k_2) \right\|_2 \preceq q^{1/2} n^{-1/2}(t_1) \log^{1/2} n(t_1), \tag{55}$$

with probability at least $1 - O(n^{-1}(t_1))$.

Moreover, with some calculations, we can show that

$$\max_{1 \le k_1 \le k_2 \le K} \left\| \widehat{V}(k_1, k_2) - \widehat{V}^*(k_1, k_2) \right\|_2 \le \sum_{a=0}^{1} \max_{j \ge 1} \| \Sigma_a^{-1} - \widehat{\Sigma}_a^{-1}(t_j) \|_2$$

$$\times \max_{1 \le k_1 \le K} \frac{2}{n(t_{k_1})} \left\| \sum_{j=1}^{k_1} \sum_{i=n(t_{j-1})+1}^{n(t_j)} \mathbb{I}(A_i = a) \varphi(X_i) \varphi^\top(X_i) \{Y_i - \varphi^\top(X_i)\widehat{\beta}_a(t_j)\}^2 \Sigma_a^{-1} \right\|_2$$

$$+ \sum_{a=0}^{1} \max_{1 \le k_1 \le K} \frac{1}{n(t_{k_1})} \left\| \sum_{j=1}^{k_1} \sum_{i=n(t_{j-1})+1}^{n(t_j)} \Sigma_a^{-1} \mathbb{I}(A_i = a) \varphi(X_i) \varphi^\top(X_i) \{Y_i - \varphi^\top(X_i)\widehat{\beta}_a(t_j)\}^2 \Sigma_a^{-1} \right\|_2$$

$$\times \max_{j \ge 1} \| \Sigma_a^{-1} - \widehat{\Sigma}_a^{-1}(t_j) \|_2^2.$$

In view of Lemma 4 and Lemma 5, we have with probability at least $1 - O(n^{-\alpha_0}(t_1))$ that

$$\max_{1 \le k_1 \le k_2 \le K} \left\| \widehat{V}(k_1, k_2) - \widehat{V}^*(k_1, k_2) \right\|_2 \le O(1)(q\delta_{n(t_1)} + \sqrt{qn^{-1}(t_1) \log n(t_1)})$$

$$\times \max_{1 \le k_1 \le K} \frac{1}{n(t_{k_1})} \left\| \underbrace{\sum_{j=1}^{k_1} \sum_{i=n(t_{j-1})+1}^{n(t_j)} \mathbb{I}(A_i = a) \varphi(X_i) \varphi^\top(X_i) \{Y_i - \varphi^\top(X_i)\widehat{\beta}_a(t_j)\}^2}_{\Psi_{3,a,k_1}} \right\|_2,$$

where $O(1)$ denotes some positive constant. Similar to equation 50 and equation 55, we can show with probability at least $1 - O(n^{-\alpha_0}(t_1))$ that

$$\max_{a \in \{0,1\}} \max_{1 \le k_1 \le K} \left\| \frac{1}{n(t_{k_1})} \Psi_{3,a,k_1} - \Psi_a \right\|_2 = o(1).$$

Similar to Lemma 4, we can show $\max_{a \in \{0,1\}} \|\Psi_a\|_2 = O(1)$. It follows that

$$\max_{1 \le k_1 \le k_2 \le K} \left\| \widehat{V}(k_1, k_2) - \widehat{V}^*(k_1, k_2) \right\|_2 \preceq q\delta_{n(t_1)} + \sqrt{qn^{-1}(t_1) \log n(t_1)},$$

with probability at least $1 - O(n^{-\alpha_0}(t_1))$. Combining this together with equation 50 and equation 55, we obtain with probability at least $1 - O(n^{-\alpha_0}(t_1))$ that

$$\max_{1 \le k_1 \le k_2 \le K} \left\| \widehat{V}(k_1, k_2) - V(k_1, k_2) \right\|_2 \preceq q\delta_{n(t_1)} + \sqrt{qn^{-1}(t_1) \log n(t_1)}.$$

Consider the function $\mathrm{g}_\delta \circ \phi_{\eta, \{\nu_k\}_k}$ defined in the proof of Theorem 1. We fix $\delta = \eta^{-1}\{\log K + 4d \log n(t_1)\}$. Based on Lemma A2 in Belloni & Oliveira (2018), we have with probability at least $1 - O(n^{-\alpha_0}(t_1))$ that

$$\sup_{\{\nu_k\}_k} \left| \mathrm{E}^* \mathrm{g}_\delta \circ \phi_{\eta, \{\nu_k\}_k}(N(0, \widehat{\boldsymbol{V}})) - \mathrm{E}\mathrm{g}_\delta \circ \phi_{\eta, \{\nu_k\}_k}(N(0, \boldsymbol{V})) \right|$$
$$\preceq q\eta^2 \{\log^2 K + \log^2 n(t_1)\} \left( q\delta_{n(t_1)} + \sqrt{qn^{-1}(t_1) \log n(t_1)} \right),$$

where $\mathrm{E}^*$ denotes the expectation conditional on the observed data. For a given set of thresholds $\{\nu_k\}_k$, using similar arguments in proving equation 31, equation 32, equation 34 and equation 35, we can show with probability at least $1 - O(n^{-\alpha_0}(t_1))$ that

$$\mathrm{Pr}^* \left\{ \max_{k \in \{1,\dots,K\}} \left( \sqrt{n(t_k)} \widehat{S}^{\mathrm{MB}*} - \nu_k \right) \le 0 \right\} \le \mathrm{E}^* \mathrm{g}_\delta \circ \phi_{\eta, \{\nu_{k,+}\}_k}(N(0, \widehat{\boldsymbol{V}}))$$

$$\le \ \mathrm{E}\mathrm{g}_\delta \circ \phi_{\eta, \{\nu_{k,+}\}_k}(N(0, \boldsymbol{V})) + O(1)q\eta^2 \{\log^2 K + \log^2 n(t_1)\} \left( q\delta_{n(t_1)} + \sqrt{qn^{-1}(t_1) \log n(t_1)} \right)$$

$$\le \ \mathrm{Pr} \left\{ \max_{k \in \{1,\dots,K\}} \left( \sup_{x \in \mathbb{X}_0} \varphi^\top(x) G(t_k) - \nu_{k,+}^* \right) \le 0 \right\}$$

$$+ \ O(1)q\eta^2 \{\log^2 K + \log^2 n(t_1)\} \left( q\delta_{n(t_1)} + \sqrt{qn^{-1}(t_1) \log n(t_1)} \right),$$

and

$$\mathrm{Pr}^* \left\{ \max_{k \in \{1,\dots,K\}} \left( \sqrt{n(t_k)} \widehat{S}^{\mathrm{MB}*} - \nu_k \right) \le 0 \right\} \ge \mathrm{E}^* \mathrm{g}_\delta \circ \phi_{\eta, \{\nu_{k,-}\}_k}(N(0, \widehat{\boldsymbol{V}}))$$

$$\ge \ \mathrm{E}\mathrm{g}_\delta \circ \phi_{\eta, \{\nu_{k,-}\}_k}(N(0, \boldsymbol{V})) - O(1)q\eta^2 \{\log^2 K + \log^2 n(t_1)\} \left( q\delta_{n(t_1)} + \sqrt{qn^{-1}(t_1) \log n(t_1)} \right)$$

$$\ge \ \mathrm{Pr} \left\{ \max_{k \in \{1,\dots,K\}} \left( \sup_{x \in \mathbb{X}_0} \varphi^\top(x) G(t_k) - \nu_{k,-}^* \right) \le 0 \right\}$$

$$- \ O(1)q\eta^2 \{\log^2 K + \log^2 n(t_1)\} \left( q\delta_{n(t_1)} + \sqrt{qn^{-1}(t_1) \log n(t_1)} \right),$$

where $O(1)$ denotes some positive constant, and

$$\nu_{k,+} = \nu_k + \eta^{-1}\{4d \log n(t_1) + \log K\} + \bar{c}^* n^{-2}(t_1), \quad \nu_{k,+}^* = \nu_{k,+} + 3\eta^{-1}\{4d \log n(t_1) + \log K\},$$
$$\nu_{k,-} = \nu_k - 3\eta^{-1}\{4d \log n(t_1) + \log K\} - \bar{c}^* n^{-2}(t_1), \quad \nu_{k,-}^* = \nu_{k,-} - \eta^{-1}\{4d \log n(t_1) + \log K\}.$$

By Theorem 2 of Chernozhukov et al. (2017), we obtain that

$$\mathrm{Pr} \left\{ \max_{k \in \{1,\dots,K\}} \left( \sup_{x \in \mathbb{X}_0} \varphi^\top(x) G(t_k) - \nu_{k,+}^* \right) \le 0 \right\} - \mathrm{Pr} \left\{ \max_{k \in \{1,\dots,K\}} \left( \sup_{x \in \mathbb{X}_0} \varphi^\top(x) G(t_k) - \nu_{k,-}^* \right) \le 0 \right\}$$

$$\preceq \eta^{-1}\{\log^{3/2} n(t_1) + \log^{3/2} K\} + \bar{c}^* n^{-2}(t_1)\{\log^{1/2} n(t_1) + \log^{1/2} K\}.$$

It follows that

$$
\sup_{\{\nu_k\}_k} \left| \Pr^* \left\{ \max_{k \in \{1,\dots,K\}} \left( \sqrt{n(t_k)} \widehat{S}^{\mathrm{MB}*} - \nu_k \right) \leq 0 \right\} - \Pr \left\{ \max_{k \in \{1,\dots,K\}} \left( \sup_{x \in \mathbb{X}} \varphi^\top(x) G(t_k) - \nu_k \right) \leq 0 \right\} \right|
$$

$$
\preceq q \eta^2 \{ \log^2 K + \log^2 n(t_1) \} \left( q \delta_{n(t_1)} + \sqrt{q n^{-1}(t_1) \log n(t_1)} \right)
$$

$$
+ \eta^{-1} \{ \log^{3/2} n(t_1) + \log^{3/2} K \} + \bar{c}^* n^{-2}(t_1) \{ \log^{1/2} n(t_1) + \log^{1/2} K \},
$$

with probability at least $1 - O(n^{-\alpha_0}(t_1))$. Set

$$
\eta = \min[ q^{-1} n^{\alpha_0/3}(t_1) \log^{-(1+2\alpha_0)/6} \{ K n(t_1) \}, q^{-1/2} n^{1/6}(t_1) \log^{-1/3} \{ K n(t_1) \} ],
$$

we obtain the desired result.

## E.8 PROOF OF LEMMA 7

Combining Lemma 6 with Lemma 4 yields that

$$
\left\| \Sigma_a^{-1} \left( \frac{1}{k} \sum_{i=1}^{k} \mathbb{I}(A_i = a) \varphi(X_i) \{ Y_i - \varphi^\top(X_i) \beta_a \} \right) \right\|_2 \preceq q^{1/2} k^{-1/2} \sqrt{\log k}, \quad \forall k \geq j_n, a \in \{0, 1\},
$$

with probability at least $1 - O(j_n^{-1})$. Combining this together with equation 16 yields that

$$
\| \widehat{\beta}_{a,k} - \beta_a \|_2 \preceq q^{1/2} k^{-1/2} \sqrt{\log k}, \quad \forall k \geq j_n, a \in \{0, 1\},
$$

with probability at least $1 - O(j_n^{-1})$. The proof is hence completed.

## E.9 PROOF OF LEMMA 8

Notice that

$$
\left\| \frac{1}{k} \sum_{i=1}^{k} \mathbb{I}(A_i = a) \varphi(X_i) \varphi^\top(X_i) (Y_i - \varphi^\top(X_i) \beta_a)^2 - \Phi_a \right\|_2
$$

$$
\leq \left\| \frac{1}{k} \sum_{i=1}^{k} \mathbb{I}(A_i = a) \varphi(X_i) \varphi^\top(X_i) [\{ Y_i - \varphi^\top(X_i) \beta_a \}^2 - \sigma^2(a, X_i)] \right\|_2
$$

$$
+ \left\| \frac{1}{k} \sum_{i=1}^{k} \mathbb{I}(A_i = a) \varphi(X_i) \varphi^\top(X_i) \sigma^2(a, X_i) - \Phi_a \right\|_2. \tag{56}
$$

Similar to the proof of Lemma 5, we can show that the second term on the RHS of equation 56 is of the order $O(q \delta_k + \sqrt{q k^{-1} \log k})$, for any $a \in \{0, 1\}$ and any $k \geq j_n$, with probability at least $1 - O(j_n^{-\alpha_0})$. As for the first term, notice that each element in the matrix

$$
\frac{1}{k} \sum_{i=1}^{k} \mathbb{I}(A_i = a) \varphi(X_i) \varphi^\top(X_i) \{ (Y_i - \varphi^\top(X_i) \beta_a)^2 - \sigma^2(a, X_i) \} \tag{57}
$$

corresponds to a martingale with respect to the filtration $\{ \sigma(\mathcal{F}_{i-1}) : i \geq 1 \}$, under (A1) and (A2). Using similar arguments in proving Equation (C.15) of Shi et al. (2020b), we can show that

$$
\left\| \frac{1}{k} \sum_{i=1}^{k} \mathbb{I}(A_i = a) \varphi(X_i) \varphi^\top(X_i) [\{ Y_i - \varphi^\top(X_i) \beta_a \}^2 - \sigma^2(a, X_i)] \right\|_2 \preceq q^{1/2} k^{-1/2} \sqrt{\log k},
$$

$$
\forall a \in \{0, 1\}, k \geq j_n,
$$

with probability at least $1 - O(j_n^{-1})$. The proof is hence completed.

### E.10 PROOF OF LEMMA 2

We begin by providing an upper bound for $\max_{a \in \{0,1\}} \|\widehat{\beta}_{a,k} - \beta_a\|_2$. With some calculations, we have

$$\max_{a \in \{0,1\}} \|\widehat{\beta}_{a,k} - \beta_a\|_2 = \max_{a \in \{0,1\}} \frac{1}{k} \left\| \widehat{\Sigma}_{a,k}^{-1} \left( \sum_{i=1}^k \varphi(X_i) \mathbb{I}(A_i = a) \{Y_i - \varphi^\top(X_i)\beta_a\} \right) \right\|_2$$

$$\leq \max_{a \in \{0,1\}} \left\| \widehat{\Sigma}_{a,k}^{-1} \right\|_2 \max_{a \in \{0,1\}} \frac{1}{k} \left\| \sum_{i=1}^k \varphi(X_i) \mathbb{I}(A_i = a) \{Y_i - \varphi^\top(X_i)\beta_a\} \right\|_2.$$

By Lemma 6, we obtain with probability at least $1 - O(j_n^{-1})$ that

$$\max_{a \in \{0,1\}} \frac{1}{k} \left\| \sum_{i=1}^k \varphi(X_i) \mathbb{I}(A_i = a) \{Y_i - \varphi^\top(X_i)\beta_a\} \right\|_2 \preceq q^{1/2} k^{-1/2} \sqrt{\log k}, \quad \forall k \geq j_n. \tag{58}$$

Similarly, we can show with probability at least $1 - O(j_n^{-1})$ that

$$\max_{a \in \{0,1\}} \frac{1}{k} \left\| \sum_{i=1}^k \varphi(X_i) \mathbb{I}(A_i = a) \{Y_i - \varphi^\top(X_i)\beta_a\} \right\|_2 \preceq q^{1/2} k^{-1/2} \sqrt{\log j_n}, \tag{59}$$

$$\forall 1 \leq k < j_n.$$

Similar to equation 42, we have

$$\max_{a \in \{0,1\}} \lambda_{\min}[\widehat{\Sigma}_{a,k}] \geq \min_{a \in \{0,1\}} \lambda_{\min} \left( \mathrm{E}^{\mathcal{F}_{i-1}} \varphi(X)\varphi^\top(X) \frac{1}{k} \sum_{i=1}^k \pi_{i-1}(a, X) \right)$$

$$- \max_{a \in \{0,1\}} \frac{1}{k} \left\| \sum_{i=1}^k \{ \mathbb{I}(A_i = a)\varphi(X_i)\varphi^\top(X_i) - \mathrm{E}^{\mathcal{F}_{i-1}} \pi_{i-1}(a, X)\varphi(X)\varphi^\top(X) \} \right\|_2.$$

Using similar arguments in proving equation 44, we can show that

$$\max_{a \in \{0,1\}} \left\| \sum_{i=1}^k \{ \mathbb{I}(A_i = a)\varphi(X_i)\varphi^\top(X_i) - \mathrm{E}^{\mathcal{F}_{i-1}} \pi_{i-1}(a, X)\varphi(X)\varphi^\top(X) \} \right\|_2 \preceq \sqrt{qk \log k}, \tag{60}$$

$$\forall k \geq j_n,$$

with probability at least $1 - O(j_n^{-1})$. Similarly, we can show

$$\max_{a \in \{0,1\}} \left\| \sum_{i=1}^k \{ \mathbb{I}(A_i = a)\varphi(X_i)\varphi^\top(X_i) - \mathrm{E}^{\mathcal{F}_{i-1}} \pi_{i-1}(a, X)\varphi(X)\varphi^\top(X) \} \right\|_2 \preceq \sqrt{qk \log j_n}, \tag{61}$$

$$\forall 1 \leq k < j_n,$$

with probability at least $1 - O(j_n^{-1})$.

Without loss of generality, assume $\varepsilon_0 \leq 1/2$. Notice that we have $\pi_{i-1}(a, x) \geq \varepsilon_0$, for any $a \in \{0,1\}$, $x \in \mathbb{X}$ and $i \geq N_0$. This together with Lemma equation 4 implies that

$$\inf_{a \in \{0,1\}, n \geq j_n} \lambda_{\min} \left( \mathrm{E}^{\mathcal{F}_{i-1}} \varphi(X)\varphi^\top(X) \frac{1}{n} \sum_{i=1}^n \pi_{i-1}(a, X) \right) \geq \frac{n - N_0}{n} \varepsilon_0 \geq \frac{j - N_0}{j} \varepsilon_0.$$

Combining this together with equation 60 and equation 61 yields

$$\max_{a \in \{0,1\}} \lambda_{\min}[\widehat{\Sigma}_{a,k}] \geq \frac{\varepsilon_0}{2}, \quad \forall k \geq L^* \sqrt{q \log j_n},$$

for some constant $L^* \geq 1$, with probability at least $1 - O(j_n^{-1})$. This together with equation 58 and equation 59 yields that

$$\max_{a \in \{0,1\}} \|\widehat{\beta}_{a,k} - \beta_a\|_2 \preceq q^{1/2} k^{-1/2} \sqrt{\log \max(k, j_n)}, \quad \forall k \geq L^* \sqrt{q \log j_n},$$

with probability at least $1 - O(j_n^{-1})$.

By Condition (A3), we have

$$|\varphi^\top(X)(\widehat{\beta}_{1,k} - \widehat{\beta}_{0,k} - \beta_1 + \beta_0)| \le \bar{L}qk^{-1/2}\log^{1/2}\max(k, j_n), \quad \forall k \ge L^*\sqrt{q\log j_n}, \quad (62)$$

for some constant $\bar{L} > 0$, with probability at least $1 - O(j_n^{-1})$.

For any $z_1, z_2 \in \mathbb{R}$, we have $\mathbb{I}(z_1 > 0) \ne \mathbb{I}(z_2 > 0)$ only when $|z_1 - z_2| \ge |z_2|$. Hence, under the event defined in equation 62, the event $\mathbb{I}\{\varphi^\top(X)(\widehat{\beta}_{1,k} - \widehat{\beta}_{0,k}) > 0\} \ne \mathbb{I}\{\varphi^\top(X)(\beta_1 - \beta_0) > 0\}$ occurs only when

$$|\varphi^\top(X)(\beta_1 - \beta_0)| \le |\varphi^\top(X)(\widehat{\beta}_{1,k} - \widehat{\beta}_{0,k} - \beta_1 + \beta_0)| \le \bar{L}qk^{-1/2}\sqrt{\log\max(k, j_n)},$$

for any $k \ge j_n$. Under the given conditions, we have

$$\Pr\left(|\varphi^\top(X)(\beta_1 - \beta_0)| \le \bar{L}qk^{-1/2}\log^{1/2}\max(k, j_n)\right) \le \bar{L}L_0 qk^{-1/2}\log^{1/2}\max(k, j_n). \quad (63)$$

Notice that when $\mathbb{I}\{\varphi^\top(X)(\widehat{\beta}_{1,k} - \widehat{\beta}_{0,k}) > 0\} = \mathbb{I}\{\varphi^\top(X)(\beta_1 - \beta_0) > 0\}$, we have $\pi_k(a, X) = \pi^*(a, X)$. Thus, we obtain $\pi_k(a, X) = \pi^*(a, X)$ if $|\varphi^\top(X)(\beta_1 - \beta_0)| > \bar{L}qk^{-1/2}\sqrt{\log\max(k, j_n)}$, for any $k \ge L^*\sqrt{q\log j_n}$. Set $k_0 = L^*\sqrt{q\log j_n}$. By equation 62 and equation 63, we have with probability at least $1 - O(j_n^{-1})$ that

$$\sum_{a\in\{0,1\}} \mathrm{E}^{\mathcal{F}_{i-1}}\left|\sum_{i=1}^k \{\pi_{i-1}(a, X) - \pi^*(a, X)\}\right| \le \sum_{a\in\{0,1\}} \sum_{i=1}^{k_0} \mathrm{E}^{\mathcal{F}_{i-1}}|\pi_{i-1}(a, X) - \pi^*(a, X)|$$

$$+ \sum_{a\in\{0,1\}} \sum_{i=k_0+1}^k \mathrm{E}^{\mathcal{F}_{i-1}}|\pi_{i-1}(a, X) - \pi^*(a, X)| \le 2L^*\sqrt{q\log j_n}$$

$$+ \sum_{a\in\{0,1\}} \sum_{i=k_0+1}^k \mathrm{E}^{\mathcal{F}_{i-1}}|\pi_{i-1}(a, X) - \pi^*(a, X)|\mathbb{I}\{|\varphi^\top(X)(\beta_1 - \beta_0)| > \bar{L}qi^{-1/2}\log^{1/2}i\}$$

$$+ \sum_{a\in\{0,1\}} \sum_{i=k_0+1}^k \mathrm{E}^{\mathcal{F}_{i-1}}|\pi_{i-1}(a, X) - \pi^*(a, X)|\mathbb{I}\{|\varphi^\top(X)(\beta_1 - \beta_0)| \le \bar{L}qi^{-1/2}\log^{1/2}i\}$$

$$\le 2L^*\sqrt{q\log j_n} + \sum_{a\in\{0,1\}} \sum_{i=k_0+1}^n \Pr\left(|\varphi^\top(X)(\beta_1 - \beta_0)| \le \bar{L}qi^{-1/2}\sqrt{\log i}\right) \preceq qk^{1/2}\log^{1/2}k, \quad \forall k \ge j_n.$$

The proof is hence completed.

## F  COMPARISON OF THE BASELINE

Consider our test statistic $S(t)$. Under $H_0$, it can be bounded from above by

$$\sup_{x\in\mathbb{X}} \varphi^\top(x)\{\widehat{\beta}_1(t) - \beta_1^* - \widehat{\beta}_0(t) + \beta_0^*\}. \quad (64)$$

It suffices to provide an upper bound for the above expression. By Cauchy-Schwarz inequality, equation 64 can be upper bounded by

$$\sup_{x\in\mathbb{X}} \|\varphi(x)\|_2\|\widehat{\beta}_1(t) - \beta_1^* - \widehat{\beta}_0(t) + \beta_0^*\|_2.$$

It suffices to provide anytime upper bound for $\|\widehat{\beta}_1(t) - \beta_1^* - \widehat{\beta}_0(t) - \beta_0^*\|_2$.

Recall that

$$\widehat{\beta}_1(t) - \beta_1^* - \widehat{\beta}_0(t) + \beta_0^*$$

$$= \frac{1}{N(t)} \sum_{i=1}^{N(t)} [\mathbb{I}(A_i = 1)\widehat{\Sigma}_1^{-1}(t)\varphi(X_i)\{Y_i - \varphi^\top(X_i)\beta_1^*\} - \mathbb{I}(A_i = 0)\widehat{\Sigma}_0^{-1}(t)\varphi(X_i)\{Y_i - \varphi^\top(X_i)\beta_0^*\}].$$

The above expression is asymptotically equivalent to

$$\frac{1}{N(t)}\sum_{i=1}^{N(t)}[\mathbb{I}(A_i=1)\Sigma_1^{-1}\varphi(X_i)\{Y_i-\varphi^\top(X_i)\beta_1^*\}-\mathbb{I}(A_i=0)\Sigma_0^{-1}\varphi(X_i)\{Y_i-\varphi^\top(X_i)\beta_0^*\}].$$

By the law of iterated logarithm, the $\ell$-th dimension of the above expression can be upper bounded by

$$N^{-1/2}(t)\sqrt{2\sigma_\ell^2\log\log\{N(t)\}},$$

where $\sum_\ell\widehat{\sigma}_\ell^2$ can be consistently estimated by

$$\frac{1}{N(t)}\sum_{i=1}^{N(t)}\|\mathbb{I}(A_i=1)\widehat{\Sigma}_1^{-1}(t)\varphi(X_i)\{Y_i-\varphi^\top(X_i)\widehat{\beta}_1(t)\}-\mathbb{I}(A_i=0)\widehat{\Sigma}_0^{-1}(t)\varphi(X_i)\{Y_i-\varphi^\top(X_i)\widehat{\beta}_0(t)\}\|_2^2.$$

As such, the finite error bound is given by

$$\sup_{x\in\mathbb{X}}\|\varphi(x)\|_2\frac{\sqrt{2\log\log\{N(t)\}}}{\sqrt{N(t)}}\times$$

$$\sqrt{\frac{1}{N(t)}\sum_{i=1}^{N(t)}\|\mathbb{I}(A_i=1)\widehat{\Sigma}_1^{-1}(t)\varphi(X_i)\{Y_i-\varphi^\top(X_i)\widehat{\beta}_1(t)\}-\mathbb{I}(A_i=0)\widehat{\Sigma}_0^{-1}(t)\varphi(X_i)\{Y_i-\varphi^\top(X_i)\widehat{\beta}_0(t)\}\|_2^2.}$$

## G    ADDITIONAL TABLES AND FIGURES

| | | method | BAT | | | | LIL | | | |
| | | | Random | | Adaptive | | Random | | Adaptive | |
| $(n,K)$ | $\delta$ | | rej probs | E[stop] | rej probs | E[stop] | rej probs | E[stop] | rej probs | E[stop] |
|---|---|---|---|---|---|---|---|---|---|---|
| | | 0.00 | 5.0(1.1) | 3537(14) | 6.2(1.2) | 3534(15) | 0.0(0.0) | 3600(0) | 0.0(0.0) | 3600(0) |
| | | 0.10 | 17.2(1.9) | 3400(24) | 18.5(1.9) | 3409(23) | 0.0(0.0) | 3600(0) | 0.0(0.0) | 3600(0) |
| | (200, 5) | 0.15 | 36.0(2.4) | 3184(32) | 35.5(2.4) | 3189(32) | 0.2(0.2) | 3599(0) | 0.0(0.0) | 3600(0) |
| | | 0.20 | 55.8(2.5) | 2914(36) | 60.0(2.4) | 2908(36) | 0.5(0.4) | 3598(1) | 0.8(0.4) | 3595(3) |
| | | 0.25 | 79.5(2.0) | 2545(35) | 81.5(1.9) | 2528(34) | 1.8(0.7) | 3590(4) | 2.8(0.8) | 3585(5) |
| S1 | | 0.30 | 93.2(1.3) | 2286(27) | 95.2(1.1) | 2280(26) | 7.2(1.3) | 3560(10) | 7.5(1.3) | 3549(11) |
| | | 0.00 | 5.2(1.1) | 3879(18) | 5.5(1.1) | 3882(18) | 0.0(0.0) | 3960(0) | 0.0(0.0) | 3960(0) |
| | | 0.10 | 17.2(1.9) | 3716(29) | 24.0(2.1) | 3651(32) | 0.0(0.0) | 3960(0) | 0.0(0.0) | 3960(0) |
| | (20, 50) | 0.15 | 39.5(2.4) | 3394(40) | 41.2(2.5) | 3365(40) | 0.0(0.0) | 3960(0) | 0.2(0.2) | 3958(1) |
| | | 0.20 | 61.8(2.4) | 3021(44) | 61.0(2.4) | 3013(44) | 0.8(0.4) | 3949(6) | 0.2(0.2) | 3956(3) |
| | | 0.25 | 84.0(1.8) | 2588(39) | 83.5(1.9) | 2579(39) | 4.8(1.1) | 3919(11) | 3.5(0.9) | 3936(8) |
| | | 0.30 | 95.8(1.0) | 2281(28) | 95.5(1.0) | 2275(28) | 12.8(1.7) | 3847(18) | 14.5(1.8) | 3873(15) |
| | | 0.00 | 5.0(1.1) | 3537(14) | 6.2(1.2) | 3534(15) | 0.0(0.0) | 3600(0) | 0.0(0.0) | 3600(0) |
| | | 0.10 | 8.8(1.4) | 3511(17) | 8.8(1.4) | 3515(16) | 0.0(0.0) | 3600(0) | 0.0(0.0) | 3600(0) |
| | (200, 5) | 0.15 | 25.5(2.2) | 3346(26) | 26.0(2.2) | 3337(27) | 0.0(0.0) | 3600(0) | 0.0(0.0) | 3600(0) |
| | | 0.20 | 57.2(2.5) | 3004(34) | 60.2(2.4) | 3000(34) | 0.0(0.0) | 3600(0) | 0.0(0.0) | 3600(0) |
| | | 0.25 | 87.2(1.7) | 2569(32) | 88.2(1.6) | 2581(32) | 2.0(0.7) | 3594(4) | 1.5(0.6) | 3589(5) |
| S1 | | 0.30 | 98.0(0.7) | 2224(21) | 97.8(0.7) | 2254(24) | 10.2(1.5) | 3559(9) | 6.2(1.2) | 3574(8) |
| | | 0.00 | 5.2(1.1) | 3879(18) | 5.5(1.1) | 3882(18) | 0.0(0.0) | 3960(0) | 0.0(0.0) | 3960(0) |
| | | 0.10 | 8.2(1.4) | 3839(21) | 7.0(1.3) | 3852(21) | 0.0(0.0) | 3960(0) | 0.0(0.0) | 3960(0) |
| | (20, 50) | 0.15 | 28.0(2.2) | 3599(34) | 27.8(2.2) | 3627(33) | 0.0(0.0) | 3960(0) | 0.2(0.2) | 3956(3) |
| | | 0.20 | 64.8(2.4) | 3165(41) | 62.0(2.4) | 3168(41) | 1.0(0.5) | 3951(5) | 0.8(0.4) | 3955(4) |
| | | 0.25 | 92.2(1.3) | 2608(35) | 90.2(1.5) | 2597(36) | 5.0(1.1) | 3926(10) | 2.8(0.8) | 3936(8) |
| | | 0.30 | 99.2(0.4) | 2238(23) | 99.2(0.4) | 2250(24) | 14.5(1.8) | 3854(16) | 13.8(1.7) | 3868(15) |

Table 1: QTE: rejection probabilities (multiplied by 100) and average stopping times under Scenarios 1 and 2 when $\alpha_1(\cdot)$ is chosen as the spending function. Standard errors are reported in the parentheses.

| | | method | BAT | | | | AVT | | | |
|---|---|---|---|---|---|---|---|---|---|---|
| | | | Random | | Adaptive | | Random | | Adaptive | |
| | $(n, K)$ | $\delta$ | rej probs | E[stop] | rej probs | E[stop] | rej probs | E[stop] | rej probs | E[stop] |
| S1 | (200, 5) | 0.00 | 5.2(1.1) | 1763(8) | 6.2(1.2) | 1762(8) | 0.2(0.2) | 1800(0) | 51.2(2.5) | 1586(11) |
| | | 0.10 | 27.5(2.2) | 1644(15) | 26.0(2.2) | 1647(14) | 6.8(1.3) | 1771(6) | 89.0(1.6) | 1344(9) |
| | | 0.15 | 45.5(2.5) | 1511(17) | 44.2(2.5) | 1527(17) | 19.0(2.0) | 1718(10) | 98.0(0.7) | 1250(6) |
| | | 0.20 | 62.5(2.4) | 1391(18) | 64.2(2.4) | 1383(18) | 42.8(2.5) | 1581(15) | 99.5(0.4) | 1196(6) |
| | | 0.25 | 80.2(2.0) | 1263(17) | 78.8(2.0) | 1266(17) | 72.8(2.2) | 1394(17) | 99.8(0.2) | 1145(5) |
| | | 0.30 | 88.2(1.6) | 1176(14) | 88.8(1.6) | 1179(14) | 89.2(1.5) | 1216(14) | 100.0(0.0) | 1091(5) |
| | (20, 50) | 0.00 | 5.8(1.2) | 1933(9) | 5.0(1.1) | 1936(9) | 0.2(0.2) | 1978(1) | 51.5(2.5) | 1621(17) |
| | | 0.10 | 27.5(2.2) | 1771(18) | 27.5(2.2) | 1771(18) | 8.0(1.4) | 1929(9) | 89.2(1.5) | 1276(13) |
| | | 0.15 | 45.5(2.5) | 1617(22) | 45.8(2.5) | 1630(21) | 25.2(2.2) | 1826(15) | 97.8(0.7) | 1166(7) |
| | | 0.20 | 67.0(2.4) | 1446(22) | 65.5(2.4) | 1459(22) | 53.8(2.5) | 1617(20) | 99.0(0.5) | 1105(6) |
| | | 0.25 | 83.8(1.8) | 1287(19) | 82.5(1.9) | 1288(19) | 79.0(2.0) | 1379(19) | 99.8(0.2) | 1061(4) |
| | | 0.30 | 92.0(1.4) | 1182(16) | 91.5(1.4) | 1193(16) | 94.0(1.2) | 1187(15) | 100.0(0.0) | 1030(2) |
| S2 | (200, 5) | 0.00 | 5.2(1.1) | 1763(8) | 6.2(1.2) | 1762(8) | 0.2(0.2) | 1800(0) | 51.2(2.5) | 1586(11) |
| | | 0.10 | 18.2(1.9) | 1692(12) | 16.8(1.9) | 1699(12) | 3.0(0.9) | 1788(3) | 82.2(1.9) | 1406(10) |
| | | 0.15 | 29.0(2.3) | 1633(15) | 25.2(2.2) | 1642(15) | 8.5(1.4) | 1762(7) | 91.8(1.4) | 1323(9) |
| | | 0.20 | 40.5(2.5) | 1559(17) | 42.0(2.5) | 1548(17) | 17.2(1.9) | 1724(10) | 97.8(0.7) | 1257(7) |
| | | 0.25 | 50.5(2.5) | 1489(18) | 49.8(2.5) | 1492(18) | 33.0(2.4) | 1641(14) | 99.0(0.5) | 1218(6) |
| | | 0.30 | 62.5(2.4) | 1407(18) | 62.5(2.4) | 1413(18) | 53.5(2.5) | 1522(16) | 99.5(0.4) | 1181(6) |
| | (20, 50) | 0.00 | 5.8(1.2) | 1933(9) | 5.0(1.1) | 1936(9) | 0.2(0.2) | 1978(1) | 51.5(2.5) | 1621(17) |
| | | 0.10 | 19.0(2.0) | 1839(16) | 19.0(2.0) | 1837(16) | 3.5(0.9) | 1961(5) | 81.0(2.0) | 1360(15) |
| | | 0.15 | 28.5(2.3) | 1763(19) | 28.0(2.2) | 1771(18) | 11.5(1.6) | 1911(10) | 90.8(1.4) | 1256(12) |
| | | 0.20 | 39.0(2.4) | 1680(21) | 41.8(2.5) | 1685(20) | 24.0(2.1) | 1830(15) | 97.5(0.8) | 1171(8) |
| | | 0.25 | 50.7(2.5) | 1592(22) | 52.5(2.5) | 1568(22) | 44.2(2.5) | 1688(19) | 99.0(0.5) | 1124(6) |
| | | 0.30 | 65.2(2.4) | 1479(22) | 63.7(2.4) | 1481(22) | 63.5(2.4) | 1539(20) | 99.2(0.4) | 1092(5) |

Table 2: ATE: rejection probabilities (multiplied by 100) and average stopping times under Scenarios 1 and 2 when $\alpha_1(\cdot)$ is chosen as the spending function. Standard errors are reported in the parentheses.

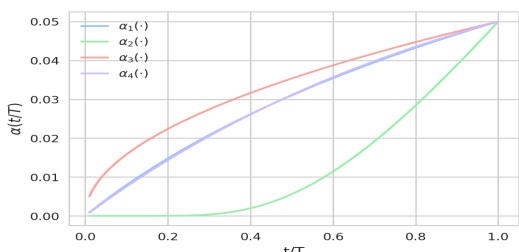

Figure 5: Alpha spending functions when $\theta = 0.5, \gamma = 1.0$.

