# OpenReview forum: "AN ONLINE SEQUENTIAL TEST FOR QUALITATIVE TREATMENT EFFECTS"
_ICLR.cc/2021/Conference — Reject_

### Official Review · AnonReviewer1 · 2020-10-27
**Official Blind Review #1**

**Rating:** 6
**Confidence:** 2

**Review:**

This paper proposed a powerful online sequential test which can efficiently detect qualitative treatment effects (QTE). The test algorithm involves adaptive randomization, sequential monitoring and online updating.  Theoretical guarantee on the Type-I error is presented.

Overall, the paper is well-written, with a clear mathematical definition of the problem, solid theoretical result and experiment results. However, there are some questions that I did not fully understand.

1. In the introduction, the difference of ATE (which is widely used in AB test in tech companies) and QTE is discussed only in words. Is there a more specific case on how QTE can be applied in tech companies where ATE fails? Could you present what are X, Y and A in your specific case, and why this case is specifically important for tech companies? I think this paper will be more interesting to readers in the industry if a detailed case is presented to bridge the introduction and the content.

2.  The author may argue that the experiment on Yahoo! Today Module is an example, but I did not find the point on this experiment. If I understand it correctly, A=0/1 represent two specific article IDs. However, in tech companies, A=0/1 usually represent two algorithms, two sets of hyperparameters or two strategies. The set of article IDs is very huge so it is not very interesting to compare the effect on two article IDs only.

---

> ### Author Response · Authors · 2020-11-23
> **Response to Reviewer 1**
>
> We greatly thank your valuable comments, many of which will lead to a much improved paper. We attempt to address all your questions one by one in the following. The revised manuscript taking into accounts all your suggestions has been uploaded. Please refer to the most updated revision for details.
>
> 1. Thanks very much for your suggestion. In addition to ATE, sometimes we are interested to locate the subgroup (if exists) that the new product performs significantly better than the existing one, as early as possible. This amounts to QTE. Take a ride-hailing company as an example. Suppose some passengers are in the recession state (at a high risk of stopping using the company’s app) and the company comes up with certain strategy to intervene the recession process. We would like to if there are some subgroups that are sensitive to our strategy and pin-point these subgroups if exists. In this case, $X$ includes age, gender and other related features, $A$ is a binary strategy indicator (whether such a strategy is applied to the passenger or not) and $Y$ is the passenger’s number of rides in the following two weeks.
>
>     We have added the related discussions in the paper (see page 1, the first paragraph in the introduction and page 2, the caption of Figure 1).
>
> 2. Thanks for your comment. We do have a dataset from a ride-hailing company comparing the performance of two subsidy strategies. However, due to privacy concerns, we cannot use that data in our paper. Therefore, we use the Yahoo! Today Module data as it is publicly available. We understand it is not very interesting to compare the effects of two articles, the dataset in our paper is used as an illustration.
>
> We once again appreciate your effort in reviewing our paper. We hope that the above discussion can address your concern.

---

### Official Review · AnonReviewer2 · 2020-10-28
**Well written paper with an interesting approach for sequential A/B testing**

**Rating:** 7
**Confidence:** 3

**Review:**

=== Contributions ===

This paper proposes a new framework for A/B testing in the frame of randomized online experiments. This new framework enables testing whether qualitative treatment effects for some specific segment(s) of the tested population can be detected or not.

The approach relies on a scalable algorithm with:
- adaptive randomization: in this setting, observations are assumed to be dependent on each other, since treatment can be adjusted by looking at previous rewards;
- a nonasymptotic upper bound on the type-I error for the online updating,
- a maximum number of data peeking times that is growing with the number of observations.

Moreover, a bootstrap method is provided in order to determine the stopping boundary. This  circumvents the absence of any tractable analytical form for the limiting distribution of this new test statistic.

Finally, the method is accompanied with experiments on the finite sample performance of the test procedure with simulated and real data from Yahoo!.

=== Strong points ===

The paper is well written and all results are well introduced with interpretation in words which makes it easy to follow.
Direct application in practice of the approach can be personalization which is currently a challenge for tech companies. Hence this paper is of great interest for the ML community.

=== Weak points ===

Minor:
Authors claim that Figure 3 reports experiment results regarding QTE. However, on this figure, we see only the results for ATE and HTE. Would it be possible to add them? I assume QTE results are better than ATE and HTE ones...

=== Recommendation ===

Overall, I vote for accepting this submission. My acceptance is supported by the strong points stated above. My grade can be further strengthened if the authors can address the points which for me need to be clarified, the biggest one being the correction of Figure 3 to fully support the efficiency of the approach.

=== Additional feedback ===

For Figure 2, what do S1 and S2 mean? I guess “random” refers to probability 0.5 to be assigned to one or the other treatment and Adaptive to the epsilon-greedy approach.

How long does it take on average to compute each test for Yahoo data? Do you recommend applying the test for low dimensional data (number of features is 5 for Yahoo)?

Minor details:
-For readability, I would add in the title of Figure 1 that A is the treatment applied and Y the associated reward, since they have not been yet introduced at the time of the figure’s reference.

For reproducibility of the results, are the Yahoo data used for the experiment freely available? If yes, would it be possible to add a link to the repository?

=== Questions to help to clarify ===

How does it relate with Bandits tests that are also online? Would it make sense to add some experiments to show when it is better to use Bandits tests over BAT method for A/B testing?

=== After authors' feedback period ===

I read carefully authors' responses to all reviews. The author's addressed my concerns and I guess the ones of the other reviewers too. One limitation that can be raised now is that the method is better suited for low-dimensional data.
Hence, I keep my accept score.

---

> ### Author Response · Authors · 2020-11-23
> **Response to Reviewer 2**
>
> We greatly thank your valuable comments, many of which will lead to a much improved paper. We attempt to address all your comments one by one in the following. The revised manuscript taking into accounts all your suggestions has been uploaded. Please refer to the most updated revision for details.
>
> **Weak points on Figure 3** Many thanks for pointing this out. We apologize for the typo. We did calculate QTE instead of HTE in the real data example and the y-axis label should be QTE instead of HTE. We have corrected this in the paper (page 8, Figure 3).
>
> **S1 and S2 in Figure 2** Thanks for the comment. We apologize for the confusion. We consider two scenarios in our simulation (page 7, line 23). S1 refers to the first scenario and S2 refers to the second scenario. As you commented, “random” refers to the first design where the treatment assignment is completely random (page 7, line 17). “Adaptive” refers to the second design where the treatment is adaptively generated (page 7, line 19). We have added these clarifications the caption of Figure 2 (page 8).
>
> **Notation in Figure 1** Thanks again for this suggestion. These notations have been introduced in the caption of Figure 1 (page 2).
>
> **Reproducibility and computation time of the real data**  The dataset is available online. Following your suggestion, we have added the link in the paper (page 7, line-1).
>
> As for the computation time, although it depends on the number of specified interim looks, on average the computation time is around several seconds. We have added the discussion in the paper (page 8, line 13)
>
> **Performance with low dimensional data** We do recommend applying the test for low dimensional data. The proposed test achieves good performance in both the synthetic and real datasets. Specifically, in our real dataset where $d=5$, we find the proposed test is consistent using A/A and A/B experiments. In our synthetic dataset where $d=3$, we find the proposed test is more powerful than other competing baselines.
>
> **Relation to the bandit tests** Thanks for this comment. In bandit tests, one typically conducts a multi-armed bandit experiment to identify the arm that receives the maximum reward. We discuss the difference and similarity between bandit tests and the proposed tests below.
>
> Difference: Similar to most of the existing A/B testing methods, it focuses on comparing the ATE between different arms whereas the proposed test mainly considers identifying the QTE. Bandit tests and the proposed tests target different problems. The two tests might not be comparable.
>
> Similarly: Both the bandit test and the proposed test apply to online experiments and allows for adaptive treatment allocation. More specifically, in bandit test, one adaptively allocates the treatment based on the observed data stream to maximize the cumulative reward. The proposed test is consistent under adaptive design as well (see Figure 2, page 8).
>
> We once again appreciate your effort in reviewing our paper. We hope that the above discussion can address your concern.

---

### Official Review · AnonReviewer3 · 2020-10-29
**Reviewer 3's Report**

**Rating:** 3
**Confidence:** 4

**Review:**

This paper studies online test for qualitative treatment tests. The authors propose a scalable online algorithm for Type 1 error control. I find the paper under-developed that the writing has to be substantially improved, and the presentation, especially in Section 3.2, is not friendly.

1. The authors claim to have an "Online" algorithm. However, I don't see that the algorithm addresses any real online challenge. The challenge of online testing is that we are doing testing at each iteration, and it is difficult to adjust all p-values. The authors didn't address this issue at all, which is a fatal flaw.

2. The linear space approximation is very artificial to me. Can the authors give some real motivating applications?

3. I highly suggest the authors to move the theorems 3 and 4 to Section 3.2, and move the derivation (which are standard) to the appendix.

Overall, I don't see much novelty, and the authors are overclaming the contribution. This is a clear rejection.

---

> ### Author Response · Authors · 2020-11-23
> **Response to Reviewer 3**
>
> We greatly thank your valuable comments, many of which will lead to a much improved paper. We attempt to address all the points one by one in the following. The revised manuscript taking into accounts all your suggestions has been uploaded. Please refer to the most updated revision for details.
>
> 1. We respectfully disagree with this comment. This comment is incorrect. We indeed addressed this issue but you may overlook this.
> Specifically, we coupled the $\alpha$-spending approach (Jennison & Turnbull, 1999) with bootstrap to adaptively adjust the p-values. The $\alpha$-spending approach allocates the total allowable type I error at each interim stage according to an error-spending function. This guarantees our test controls the type-I error. As a result, we show both theoretically (Theorem 4, page 6) and numerically (Figure 2, page 8) that our test is valid.
>
>     Please refer to page 2, line 12 and page 6, line 14 for details.
>
> 2. First, we did not require the approximation error to be zero. The proposed test is valid as long as the approximation error converges at certain rates. In Appendix B, we discuss the approximation space that satisfies these conditions.
>
>     Second, the linear approximation space is used to facilitate the computation. In online experiments, the decision is made every few minutes to determine whether to stop the experiment or continue collecting more data. Nonlinear models such as neural networks, on one hand, are much more computationally expensive. On the other hand, it is difficult to quantify the uncertainty of neural network estimates and derive the corresponding p-value.
>
>     Third, unlike complicated nonlinear models, the linear approximation space makes the estimated Q-function more stable and interpretable. This is important in industrial applications.
>
> 3. Following your suggestion, we have moved the two theorems in the main text (page 6).
>
> We once again appreciate your effort in reviewing our paper. We hope that the above discussion can address your concern.

---

### Official Review · AnonReviewer4 · 2020-10-29
**online sequential test**

**Rating:** 4
**Confidence:** 1

**Review:**

I am a statistician but I am not an expert in sequential test. My questions and remarks can therefore only be those of a rather naive reader. I understand the motivation behind the paper, but I would clearly have liked to be able to clearly understand the assumptions used. I would also have to work on a  "real" algorithm for which one is able to control "everything" (in particular, the strategy to choose the . I do not have the impression that this is completely the case here and some aspects need to be clarified.

- I do not understand Eq. (1). The null and the alternative in this case depends on $\beta_0,\beta_1$ this is very strange.
- I do not understand (A2): since $\{X_k, Y_k^*(0), Y_k^*(1)\}_{k \geq 1}$ are independent, it suffices to say that $A_i$ is measurable with respect to $ \mathcal{F}_i \vee \sigma(X_i)$
- Theorem~1: the key is to check Eq. (5) which means that you have a strategy to choose the arms with a non trivial regret bound ?  What are the typical values of $\alpha_0$ ? How this is related to contextual bandits (since there is notion regret), to best arm identification problems in linear bandits ?
- The meaning of Eqs. (6) and (7) are very difficult to grasp
- How your results compare to multi-armed bandit testing with online FDR control ?

---

> ### Author Response · Authors · 2020-11-23
> **Response to Reviewer 4**
>
> We greatly thank your valuable comments, many of which will lead to a much improved paper. In particular, we have added discussions on the assumption. In our numerical experiments, we also have a setting where the treatments are adaptively generated according to the $\epsilon$-greedy policy in order to balance the tradeoff between exploration and exploitation (see the results in Figure 2 under adaptive designs). We attempt to address all the points one by one in the following. The revised manuscript taking into accounts all your suggestions has been uploaded. Please refer to the most updated revision for details.
>
> **Equation (1)** We apologize for the confusion. It should be $\beta_0^*$ and $\beta_1^*$. They are defined as the true parameters in the Q-function (page 3, line 5). We have corrected the notations in the paper (Equation (1), page 3).
>
> **Assumption (A2)** (A2) is different from the assumption that $A_i$ is measurable with respect to $\mathcal{F_{i-1}}\cup \sigma(X_i)$. Take a completely randomized study as an example where ${A_1,A_2,\cdots,}$ is independent of ${(X_1,Y_1), (X_2,Y_2),\cdots}$. In this case, (A2) is satisfied. However, $A_i$ is not a deterministic function of $\mathcal{F_{i-1}}$ and $\sigma(X_i)$. Consequently, the assumption that $A_i$ is measurable with respect to $\mathcal{F_{i-1}}\cup \sigma(X_i)$ is violated.
>
> In the literature, Assumption (A2) is referred to as the sequential randomization assumption (Zhang et al., 2013). It essentially assumes there is no unmeasured confounders and is automatically satisfied in a randomized study where the treatments are independently generated of the observed data. It is also satisfied when treatments are adaptively generated according to $\epsilon$-greedy, upper confidence bound or Thompson sampling algorithms. It guarantees that the causal estimand (defined through the potential outcomes) is estimable from the observed dataset. We have added the related discussions in the paper (page 3, line 12).
>
> **Theorem 1**. Thanks for your comment! We first clarify the meaning of Equation (5). It requires the strategy to choose the arms aggregated over different decision points to converge to a fixed strategy under certain rate.
>
> We next discuss on the choice of $\alpha_0$. In Appendix C, we show the parameter $\alpha_0=1/2$ when an $\epsilon$-greedy strategy is used for randomization to balance the trade-off between exploration and exploitation. Please see Page 12 for details.
>
> Finally, we discuss the relation to contextual bandit. As we have commented, this condition holds with $\alpha_0=1/2$ when an $\epsilon$-greedy strategy is used. More generally, one might use other commonly-employed randomizations strategies in the linear bandits (e.g., UCB or Thompson sampling). This condition holds as long as the randomization strategy converges at certain rate. We have added the related discussions in the paper (page 5, line 4).
>
> **Equations (6) and (7)** In Appendix C, we show the parameter $\alpha_0=1/2$ when an $\epsilon$-greedy strategy is used for randomization to balance the trade-off between exploration and exploitation. (6) is thus equivalent to require the number of basis function $q$ to grow at a rate slower than $N^{1/6}(t_1)$. It is automatically satisfied when $q$ is bounded. (7) is satisfied when $K$ grows polynominally fast with respect to $n$.
>
> We have added these interpretations in the page (page 5, line 1).
>
> **Online FDR control** Thanks again for this comment! Our problem is different from multi-armed bandit testing with online FDR control and is thus not comparable.
>
> First, we consider a single null hypothesis in our paper whereas multi-armed bandit testing with online FDR control consider settings with multiple testing hypotheses.
>
> Second, for the multi-armed bandit testing problem, the goal is to control FDR whereas in our setup, we aim to control the type-I error of the test procedure.
>
> Finally, we remark that it is also interesting to extend our current proposal to settings with multiple treatments. This yields a multiple testing problem with online FDR control. We leave this for future research. We have added the related discussions in the paper (Section 5, page 8).
>
> We once again appreciate your effort in reviewing our paper. We hope that the above discussion can address your concern.

---

### Decision · Program_Chairs · 2021-01-07
**Final Decision**

**Decision:**

Reject

**Comment:**

The paper proposes a new framework for online hypothesis testing aimed at detecting causal effects (of treatments on outcomes) within subgroups in online settings where treatments are randomized.  Such settings occur in online advertising where different versions of the same website may be presented to a set of otherwise exchangeable users via A/B testing.

Under the standard causal assumptions of SUTVA, and sequential ignorability, in addition to a set of regularity conditions, the authors derive a result (Theorem 1) leading to an online test (Theorem 2).  Since the resulting test's limiting distribution does not have an exact analytic form, the authors instead propose a bootstrap approach to determine a set of parameters to properly control the error rate.

The author validate their approach by a simulation study, as well as via a user click log data from Yahoo!

The reviewer opinion was somewhat split on this paper, in particular some reviewers raised concern about some (conceptually significant) typos, interpretability of assumptions, and the need for parametric assumptions (the dichotomy between linear models and neural networks is surely a false one -- the semi-parametric literature obtains nice parametric style results, although perhaps not always for tests, without assuming parametric likelihoods all the time).